# Surpassing millisecond coherence in on chip superconducting quantum memories by optimizing materials and circuit design

Suhas Ganjam [1,2] ✉, Yanhao Wang [1,2], Yao Lu [1,2], Archan Banerjee[1,2], Chan U Lei[1,2], Lev Krayzman [1,2], Kim Kisslinger[3], Chenyu Zhou [3], Ruoshui Li[3], Yichen Jia[3], Mingzhao Liu [3], Luigi Frunzio [1,2] & Robert J. Schoelkopf[1,2] ✉

The performance of superconducting quantum circuits for quantum computing has advanced tremendously in recent decades; however, a comprehensive understanding of relaxation mechanisms does not yet exist. In this work, we utilize a multimode approach to characterizing energy losses in superconducting quantum circuits, with the goals of predicting device performance and improving coherence through materials, process, and circuit design optimization. Using this approach, we measure significant reductions in surface and bulk dielectric losses by employing a tantalum-based materials platform and annealed sapphire substrates. With this knowledge we predict the relaxation times of aluminum- and tantalum-based transmon qubits, and find that they are consistent with experimental results. We additionally optimize device geometry to maximize coherence within a coaxial tunnel architecture, and realize on-chip quantum memories with single-photon Ramsey times of 2.0 – 2.7 ms, limited by their energy relaxation times of 1.0 – 1.4 ms. These results demonstrate an advancement towards a more modular and compact coaxial circuit architecture for bosonic qubits with reproducibly high coherence.

The emergence of superconducting qubits as a promising platform for quantum computing has been facilitated by over two decades of steady improvements to coherence and gate fidelity[1]. This has enabled the demonstration of many milestones, including quantum error correction or mitigation[2–9], quantum algorithms[10,11], quantum simulations[12–15], and quantum supremacy[16] using large numbers of qubits. However, the realization of a practical quantum computer requires far higher gate fidelities[17,18], which necessitate further mitigation of decoherence mechanisms in quantum circuits. Substantial exploration in the past has shown that the sources of decoherence can be traced to intrinsic sources of energy loss from the circuits' constituent materials and has revealed the existence of significant bulk and surface dielectric loss[19–23], two-level-system (TLS) loss[24–28], and

residual quasiparticle or vortex loss in superconductors[29–33]. Accordingly, improvements to coherence have been made by using intrinsically lower-loss materials such as sapphire substrates[34,35], and tantalum thin films[36,37]; and employing contamination-minimizing fabrication processes such as acid-based etching[30,33], substrate annealing[22,37,38], and thin-film encapsulation[39]. Additionally, dramatic improvements have also been achieved by modifying circuit geometry to reduce sensitivity to loss, an approach that has given rise to 3D transmon qubits[40] and cavity-based quantum memories with millisecond coherence times[41–43].

Improving coherence requires understanding the underlying loss mechanisms. Determining where the dominant losses originate as well as the extent to which those losses dominate is crucial to maximizing

[1]Departments of Applied Physics and Physics, Yale University, New Haven 06511 CT, USA. [2]Yale Quantum Institute, Yale University, New Haven 06511 CT, USA. [3]Center for Functional Nanomaterials, Brookhaven National Laboratory, Upton 11973 NY, USA. ✉e-mail: suhasganjam@google.com; robert.schoelkopf@yale.edu

the performance of superconducting qubits. There have been significant efforts to understand and mitigate surface dielectric loss in thin-film resonators[38,44–47]; however, recent studies have shown that bulk dielectric loss can play a significant role[48,49]. A systematic approach is therefore desired to characterize intrinsic losses and improve coherence in a predictable way.

Conveniently, superconducting microwave resonators are powerful characterization tools because they can be measured easily with high precision, and their quality factors are limited by the same intrinsic sources of loss as transmon qubits[33]. Additionally, their sensitivities to particular sources of loss can be tuned by modifying their geometries, a feature that has been heavily utilized in other studies to investigate various sources of loss in thin-film resonators and bulk superconductors[33,38,45–47,50–52]. In a multimode approach to loss characterization, a single device can have multiple resonance modes that are each sensitive to different sources of loss. This allows for the use of a single device to study multiple sources of loss, eliminating systematic errors due to device-to-device or run-to-run variation[33]. Furthermore, by measuring multiple multimode devices, the device-to-device variation of intrinsic loss can be determined, allowing for the evaluation of the consistency of a particular materials system or fabrication process and the prediction of the expected energy relaxation rate of a quantum circuit.

In this work, we introduce the tripole stripline, a multimode superconducting microwave resonator whose modes can be used to distinguish between surface losses, bulk dielectric loss, and package losses in thin-film superconducting quantum circuits. We use this loss characterization device to measure and compare the losses associated with thin-film aluminum and tantalum deposited on a variety of sapphire substrates that differ by their growth method and preparation. While previous work has shown improved device coherence using tantalum-based fabrication processes[36,38] and annealed sapphire substrates[37,53], we use our technique to show that the aforementioned improvements originate definitively from the reduction of surface loss in tantalum-based devices and of bulk dielectric loss in annealed sapphire substrates.

With the tripole stripline, we gain a comprehensive understanding of how materials and fabrication processes limit the coherence of superconducting quantum circuits. We use this knowledge to predictively model the loss of aluminum- and tantalum-based transmon qubits. We then confirm through transmon coherence measurements that the reduction of surface loss yielded by a tantalum-based process

results in a $T_1$ improvement of a factor of two in tantalum-based transmons over aluminum-based transmons. Understanding the loss mechanisms that limit coherence informs optimization and circuit design choices to further improve device coherence. We optimize device geometry to maximize coherence in a particular coaxial architecture, and design a stripline-based quantum memory with coherence times exceeding one millisecond. This far surpasses those of previous implementations of thin-film quantum memories[20,54], and enables the miniaturization of highly coherent bosonic qubits within larger multiqubit systems for quantum information processing.

## Results

### Characterizing microwave losses in thin films with tripole striplines

Differentiating between the various sources of loss in superconducting quantum circuits requires an appropriately designed loss characterization system. We implement such a system in the coaxial tunnel architecture[20] using multimode thin-film stripline resonators fabricated on sapphire substrates. The devices are inserted into a cylindrical tunnel waveguide package made of conventionally machined high-purity (5N5) aluminum (Fig. 1a). End-caps close the tunnel, creating a superconducting enclosure with well-defined package modes that are high (>18 GHz) in frequency (see Methods "Device packaging").

We design the multimode tripole stripline to distinguish between package losses due to induced current flowing in dissipative regions of the cylindrical tunnel package, bulk dielectric loss in the substrate, and surface dielectric losses associated with the various interfaces between substrate, superconductor, and air/vacuum. The tripole stripline is comprised of three superconducting strips placed adjacently to each other on a substrate with different widths and spacings (Fig. 1b). The arrangement of the three strips affects the spatial distributions of the electromagnetic fields of the three fundamental modes, thereby determining their sensitivities to particular sources of loss. The D1 differential mode is highly sensitive to surface losses due to its spatially localized electromagnetic field in the small 10 μm spacing between the 10 μm narrow strip and the adjacent 400 μm wide strip. On the other hand, the large 1.2 mm spacing between the two wide strips supports the D2 differential mode, whose fields are much more dilute, resulting in lower surface loss while still retaining large sensitivity to bulk dielectric loss. Finally, the common (C) mode supports a spatially diffuse electromagnetic field that induces larger electromagnetic fields

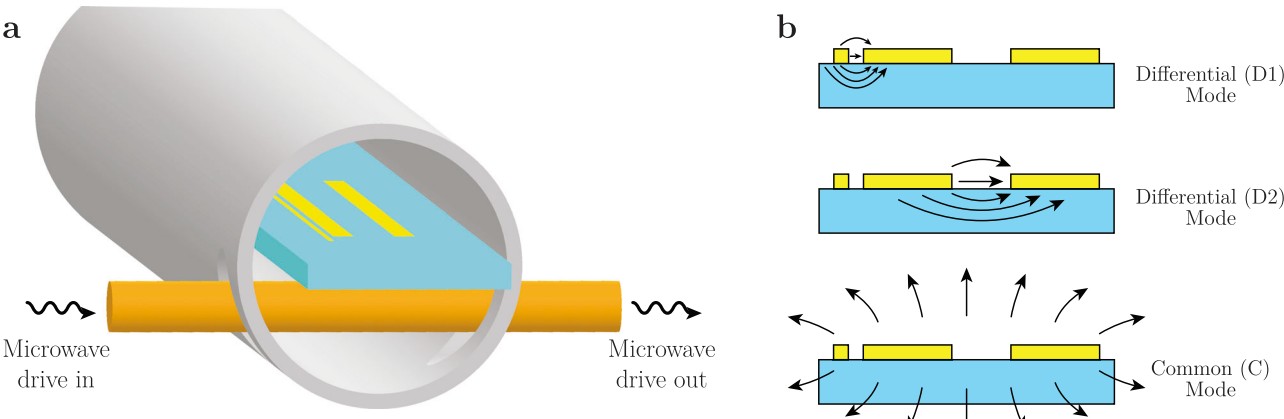

**Fig. 1 | Tripole striplines in the coaxial tunnel architecture. a** Superconducting thin-film strips are patterned on a substrate and loaded into a cylindrical tunnel made of high-purity aluminum. Resonator frequencies range from 4–7 GHz (see Supplementary Table S5). A transversely oriented coupling pin is used to capacitively drive the resonators in a hanger configuration. **b** Cross-sectional view of the tripole stripline, showing the arrangement of the strips and electric field behaviors for each mode. While the electric field of the D1 mode is confined mostly on the surface, the electric field of the D2 mode penetrates far deeper into the bulk, rendering it sensitive to losses over a significant portion of the bulk of the substrate.

on the walls of the package, rendering this mode sensitive to package loss. The differential sensitivity of these modes to different sources of loss allows us to distinguish between them by measuring the mode quality factors.

Losses in the tripole stripline can be described using a generalized energy participation ratio model[25,44,55]:

$$\frac{1}{Q_{\mathrm{int}}} = \frac{1}{\omega T_1} = \sum_i \frac{1}{Q_i} = \sum_i p_i \Gamma_i, \qquad (1)$$

where $Q_{\mathrm{int}}$ is the total internal quality factor of the resonator, $\omega$ is the resonance frequency, and $T_1$ is the energy decay time. The total loss can be broken down into a sum of losses $1/Q_i$ from distinct loss channels, where $\Gamma_i$ is the generalized intrinsic loss factor[45] associated with the $i$th loss channel, and $p_i = U_i/U_{\mathrm{tot}}$ is the geometric energy participation ratio calculated by computing the fraction of energy stored in the $i$th lossy region when a resonance mode is excited. The participation ratio is therefore determined by the spatial distribution of the electromagnetic field of the resonance mode and, as a result, can be calculated in finite-element simulation and engineered to alter the mode's sensitivity to specific loss channels (see Methods "Calculation of participation ratios"). The loss factors, on the other hand, are intrinsic material- and process-dependent quantities such as loss tangents and surface resistances that must be measured.

We use the participation ratio model in order to quantify the losses in the tripole stripline (see Supplementary Table S5). We define surface loss as $1/Q_{\mathrm{surf}} = p_{\mathrm{surf}}\Gamma_{\mathrm{surf}}$, where $p_{\mathrm{surf}} = p_{\mathrm{SA}} + p_{\mathrm{MS}} + p_{\mathrm{MA}}$ is the sum of surface dielectric participations in thin (3 nm) dielectric (relative permittivity $\epsilon_r = 10$) regions located at the substrate-air (SA), metal-substrate (MS), and metal-air (MA) interfaces. $\Gamma_{\mathrm{surf}}$ is the corresponding surface loss factor that describes the intrinsic loss in these three interrelated regions. This formulation of surface loss differs from that of other studies[45-47], which aim to independently characterize the surface loss factors $\Gamma_{\mathrm{SA}}$, $\Gamma_{\mathrm{MS}}$, and $\Gamma_{\mathrm{MA}}$; here, $\Gamma_{\mathrm{surf}}$ is a weighted sum of the three surface loss factors and characterizes the overall surface loss due to the presence of oxides, amorphous species, interdiffusion, organic residues, point-like defects, or lattice distortions. In the coaxial tunnel architecture, the three surface participation ratios retain roughly the same relative proportions regardless of circuit geometry or field distribution; therefore, $\Gamma_{\mathrm{surf}}$ becomes effectively geometry-independent. Furthermore, because the aforementioned physical signatures of loss are heavily influenced by processes such as substrate preparation, metal deposition, and circuit patterning, the three surface loss factors are interdependent; therefore, $\Gamma_{\mathrm{surf}}$ is the most relevant descriptor of intrinsic surface loss because it characterizes a particular materials platform and fabrication process in order to predict the total surface loss in a device.

We consider the surface loss to be distinct from the bulk loss $1/Q_{\mathrm{bulk}} = p_{\mathrm{bulk}}\Gamma_{\mathrm{bulk}}$ which is dielectric in nature and may be dependent on the crystalline order of the substrate. Additionally, we define package losses $1/Q_{\mathrm{pkg}} = p_{\mathrm{pkg_{cond}}}\Gamma_{\mathrm{pkg_{cond}}} + p_{\mathrm{pkg_{MA}}}\Gamma_{\mathrm{pkg_{MA}}} + p_{\mathrm{seam}}\Gamma_{\mathrm{seam}}$ as a combination of conductor loss due to residual quasiparticles, MA surface dielectric loss due to the metal oxide on the surface of the tunnel package, and seam loss $p_{\mathrm{seam}}\Gamma_{\mathrm{seam}} = y_{\mathrm{seam}}/g_{\mathrm{seam}}$ due to a contact resistance that manifests when two metals come into contact (see Methods "Calculation of participation ratios"), which occurs when the tunnel package is closed with the end-caps[33,50]. The large $p_{\mathrm{surf}}$ in the tripole stripline's D1 mode and large $p_{\mathrm{pkg_{cond}}}$, $p_{\mathrm{pkg_{MA}}}$, and $y_{\mathrm{seam}}$ in the C mode yields a participation matrix that is well-conditioned to extract the loss factors with minimal error propagation, a crucial requirement for characterizing microwave losses in this way.

## Extracting intrinsic loss factors from resonator measurements

We demonstrate loss characterization by fabricating and measuring tripole stripline resonators. Tripole striplines made of tantalum were fabricated on a HEMEX-grade sapphire substrate. The substrate was annealed at 1200 °C in oxygen before tantalum was deposited via DC magnetron sputtering at 800 °C. The striplines were patterned using a subtractive process (see Methods "Device fabrication") and then loaded into multiplexed coaxial tunnel packages (see Methods "Device packaging") and measured in hanger configuration in a dilution refrigerator at a base temperature of 20 mK. The frequency response of each mode was measured using a vector network analyzer, and the internal quality factor as a function of the average circulating photon number $\bar{n}$ was extracted by fitting the resonance circle in the complex plane (see Methods "Measurement of resonator quality factor")[56].

The differences in power dependence of $Q_{\mathrm{int}}$ of the tripole stripline's modes reflect the modes' sensitivities to surface loss (Fig. 2a). The D1 mode has the largest power dependence, with $Q_{\mathrm{int}}$ changing by over an order of magnitude from one to one million photons circulating in the resonator. We attribute this power dependence to the presence of anomalous two-level systems (TLSs) that couple to the electric field of the mode and provide additional pathways for energy relaxation to occur. Beyond a critical photon number, the TLSs become saturated and effectively decouple from the mode, causing $Q_{\mathrm{int}}$ to increase. The power dependence of each mode is fit to the following TLS model:

$$\frac{1}{Q_{\mathrm{int}}} = \frac{1}{Q_0} + \frac{p_{\mathrm{surf}}\tan\delta_{\mathrm{TLS}}}{\sqrt{1 + (\bar{n}/n_c)^\beta}}, \qquad (2)$$

where $1/Q_0$ is the power-independent contribution to the total internal loss, $\tan\delta_{\mathrm{TLS}}$ is the ensemble TLS loss tangent, $n_c$ is the critical photon number beyond which the TLSs begin to saturate, and $\beta$ is an empirical parameter that describes TLS saturation[24-26,38,57,58]. While the D2 and C modes are also power-dependent, they are far less so, with $Q_{\mathrm{int}}$ changing by less than a factor of two over the same range of $\bar{n}$. This is consistent with these modes having nearly two orders of magnitude smaller surface participation, which allows the D2 and C modes to attain single-photon $Q_{\mathrm{int}}$ that are over an order of magnitude higher than that of the D1 mode. The D2 mode, being relatively insensitive to both surface and package losses, has a single-photon $Q_{\mathrm{int}}$ of around $3 \times 10^7$, which, to our knowledge, far exceeds the highest single-photon $Q_{\mathrm{int}}$ measured in a lithographically-patterned thin-film resonator to date.

To extract the intrinsic loss factors and distinguish them from the geometric contribution to the total loss, we use the participation ratio model to define a linear system of equations $\kappa_j = \sum_i P_{ji}\Gamma_i$, where $\kappa_j = 1/Q_j$ for the $j$th mode of the tripole stripline, and $P_{ji}$ is the participation matrix (see Supplementary Table S5) of the loss characterization system. The problem reduces to solving a matrix equation using a least-squares algorithm with solution $\vec{\Gamma} = \mathbf{C}\tilde{\mathbf{P}}^{\mathrm{T}}\tilde{\vec{\kappa}}$ [59], where $\tilde{P}_{ji} = P_{ji}/\sigma_{\kappa_j}$ and $\tilde{\kappa}_j = \kappa_j/\sigma_{\kappa_j}$ are the measurement-error-weighted participation matrix and internal loss, respectively, and $\mathbf{C} = (\tilde{\mathbf{P}}^{\mathrm{T}}\tilde{\mathbf{P}})^{-1}$ is the covariance matrix. The measurement uncertainty $\sigma_{\kappa_j}$ of the internal loss $\kappa_j$ propagates onto the uncertainty of the extracted loss factor as $\sigma_{\Gamma_i} = \sqrt{C_{ii}}$ (see Methods "Extraction of loss factors using least-squares minimization"). We use the TLS model from Eq. (2) as an interpolating function to determine $Q_{\mathrm{int}}$ at all values of $\bar{n}$ and extract the intrinsic surface, bulk, and package-seam loss factors as a function of $\bar{n}$ using the least-squares algorithm (Fig. 2b). The contributions of conductor and MA surface losses from the package were calculated using previously measured loss factors for 5N5 aluminum (see Methods "Subtraction of package conductor and dielectric losses").

Mapping the mode quality factors to geometry-independent loss factors in this way allows us to observe general trends in different

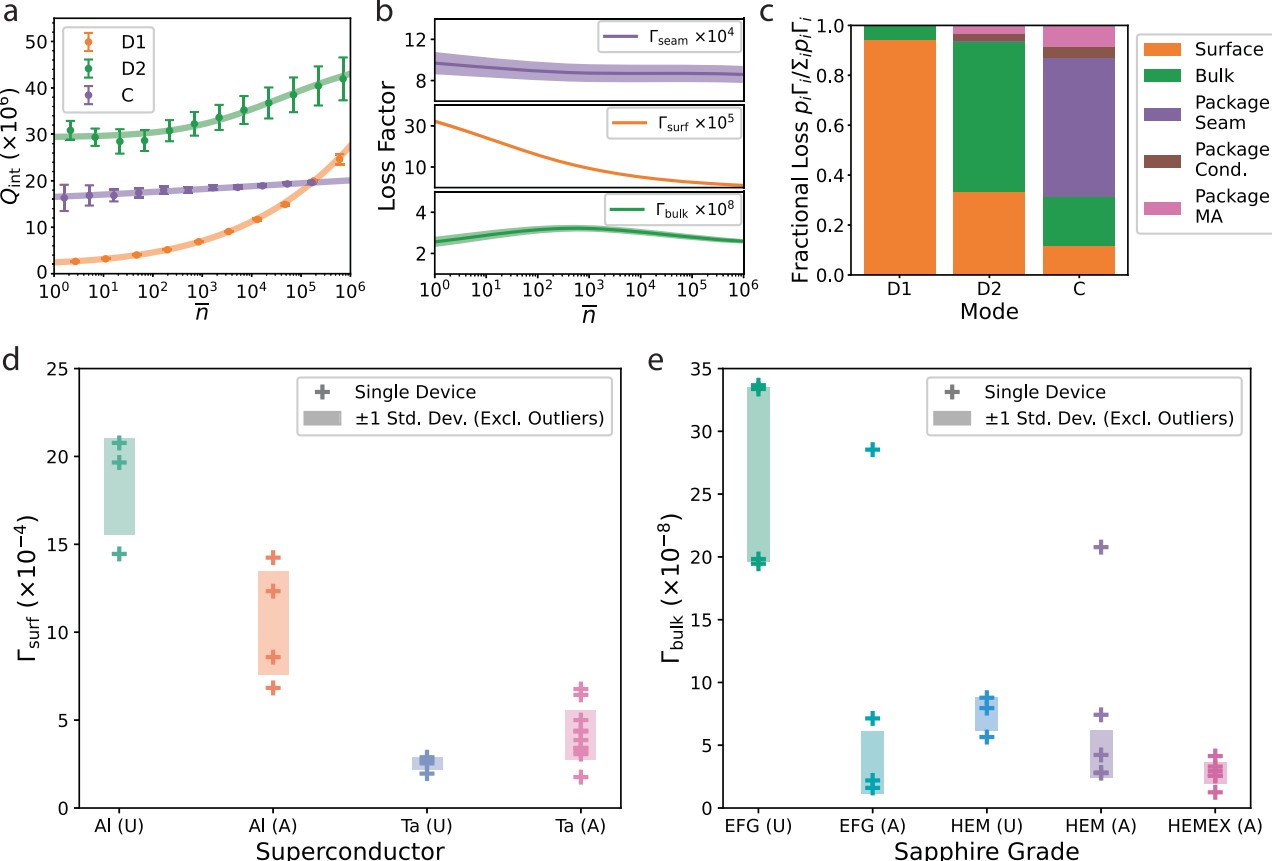

**Fig. 2 | Extraction of intrinsic losses with the tripole stripline. a** Power dependence of internal quality factor of the modes of a particular tripole stripline device, made using tantalum patterned on an annealed HEMEX sapphire substrate. Circles are measured $Q_{int}$; lines are TLS fits using Eq. (2). Error bars represent the propagated fit error on $Q_{int}$ obtained from least-squares minimization of Eq. (9) and for some points they are small enough to not be visible. The coupling quality factors $Q_c$ for this device are $6.3 \times 10^6$, $2.2 \times 10^6$, and $2.0 \times 10^6$ for the D1, D2, and C modes, respectively. The relatively large error bars on the measured $Q_{int}$ of the D2 and C modes (fractional errors of 7 and 17%, respectively, at single-photon powers) can be attributed to these modes being in the overcoupled regime and reduced signal-to-noise ratio at low excitation powers. **b** Power dependence of extracted loss factors (solid lines). The propagated error (shaded regions) for $\Gamma_{surf}$ is small (~3%) and is

hidden within the width of the solid line. Seam loss here has been normalized to be dimensionless, $\Gamma_{seam} = \omega\epsilon_0/g_{seam}$. $\Gamma_{bulk}$ slightly increases at intermediate photon numbers; we hypothesize that spatial inhomogeneities in TLS saturation within a single device could lead to the appearance of non-monotonicity in the extracted loss factor. **c** Single-photon loss budget for the modes of the tripole stripline. While the D1 mode is clearly dominated by surface loss, the D2 mode is dominated by bulk dielectric loss, and the C mode is dominated by seam loss. **d, e** Comparison of surface (**d**) and bulk (**e**) loss factors from multiple tripole stripline devices made using either aluminum- or tantalum-based fabrication processes on annealed (A) or unannealed (U) sapphire substrates. The device-to-device variation here captures the spatial inhomogeneity of the loss factors and their TLS properties.

sources of loss. We see that the surface-dependent D1 mode is power-dependent while the others are significantly less so. This implies that the surface loss factor is power-dependent while the other loss factors are not, and that the small power dependence of the D2 and C modes stem from their small but nonzero surface participation. Indeed, this is confirmed in Fig. 2b, where we observe the extracted surface loss factor is heavily power-dependent in sharp contrast with the bulk and seam loss factors. The relative power independence of the bulk and package loss factors also implies that the TLSs that dominantly couple to superconducting microwave resonators are localized in surface dielectric regions[25]. The distinction between surface and bulk dielectric loss is also apparent in the several orders of magnitude difference between the corresponding loss factors. We extract a single-photon bulk loss factor of $(2.6 \pm 0.2) \times 10^{-8}$, while the extracted single-photon surface loss factor is nearly four orders of magnitude higher at $(3.4 \pm 0.3) \times 10^{-4}$, which is qualitatively similar to what is observed in other studies[38,49,60].

To quantify the extent to which each mode is limited to a particular source of loss, we calculate a single-photon loss budget by plotting the fractional loss contribution $p_i\Gamma_i/\sum_i p_i\Gamma_i$ of each source of loss for each mode in Fig. 2c. The loss budget for the three modes

shows that the tripole stripline fulfills the ideal conditions for a loss characterization system: each mode's $Q_{int}$ is dominated by a different source of loss.

To measure how the choice of sapphire grade, wafer annealing treatment, and superconducting thin-film process affects the bulk and surface loss factors (Fig. 2d, e), we repeat the multimode approach for a variety of materials and process combinations. Multiple devices were measured for each set of materials and fabrication processes to capture the device-to-device variation of loss factors. We remark that while some outliers exist, the majority of the data points for each materials and process combination are well clustered; average and standard deviation of the loss factors are calculated excluding outliers with median relative deviation greater than 3 (see Supplementary Table S3).

We find that surface loss factors can be highly dependent on initial substrate treatment, type of superconductor, and lithography process. Aluminum-based fabrication processes on unannealed substrates yield the largest surface loss factors, while annealing the substrate improves the surface losses by a factor of two. However, the tantalum-based fabrication process yields over a factor of 2 reduction in surface loss when compared to the best aluminum-based process regardless of

whether the substrate was annealed. Cross-sectional transmission electron microscopy (TEM) of aluminum- and tantalum-based devices revealed marked differences in the MS interface. Whereas the aluminum films had a thin, ≈2-nm-thick amorphous region between the metal and the substrate, the tantalum films had a clean interface with nearly epitaxial film growth and no observable sign of an amorphous region (see Supplementary Note 8: "TEM film characterization"). It should be noted that the aluminum-based devices were deposited using electron-beam evaporation and patterned using a liftoff process while the tantalum-based devices were deposited via high-temperature sputtering and patterned using a subtractive process (see Methods "Device fabrication"). Therefore, the effects of these processes on surface quality must be considered as a convolution of the materials used and the fabrication processes employed. The differences in surface quality of aluminum- and tantalum-based thin films may be due to differences in deposition conditions, lithographic patterning, or materials compatibility with the substrate, all of which can influence how the film grows on the substrate[61,62].

Extracted bulk loss factors also vary based on the choice of sapphire grade and annealing treatment. We find that annealing EFG- and HEM-grade sapphire results in reductions in bulk dielectric loss by factors of ~8 and 2, respectively. Additionally, annealing HEMEX-grade sapphire yields the lowest bulk loss with the smallest amount of device-to-device variation as measured over six devices. The improvement through annealing is correlated with improved surface morphology observed through atomic force microscopy (AFM), which revealed atomically-flat surfaces with a monatomically-stepped terrace structure after annealing (see Supplementary Note 6: "Sapphire annealing"). It should be noted that while the difference between unannealed EFG and HEM is in qualitative agreement with other studies[49,63], the absolute bulk loss tangents differ significantly. This discrepancy can be due to the effects of the substrate undergoing the fabrication process. The samples in ref. 49 were cleaned, cleaved, and measured with no further processing. Our measurements were taken after the substrate had been through the entire fabrication process; most notably, the wafer was diced, which is a more violent process that causes chipping of the sapphire at the edges and may cause more subsurface damage that could affect the bulk loss factor.

Finally, while we observe moderate device-to-device variation in surface and bulk loss factors, we observe the extracted seam losses to vary by over two orders of magnitude over multiple nominally identically prepared cylindrical tunnel packages (Supplementary Table S2).

Device-to-device variation in interface quality due to residual contamination, interface roughness, and clamping force can result in large variations in seam conductance. This highlights the significance of package losses in the coaxial architecture as a potential source of large device-to-device variation in $Q_{int}$. However, tripole striplines are capable of characterizing this variation due to the seam loss-sensitivity of the common mode. Moreover, the high-Q modes in this section and in future sections are designed to be insensitive to seam loss, rendering it a relatively insignificant contributor to the total internal loss. We can nevertheless calculate an expected seam conductance per unit length $g_{seam} = (2.1 \pm 2.0) \times 10^2 (\Omega m)^{-1}$ by excluding outliers with a large relative deviation from the median (see Supplementary Table S2); the large uncertainty on this value is a reflection on the intrinsic variation we should expect in a device made using this particular architecture.

## Validating the loss model with qubit measurements

Microwave loss characterization is useful insofar as it can be applied to understand the losses of a candidate device of the desired geometry. We demonstrate the utility of our loss characterization technique by using the extracted loss factors from the previous section to predict the internal quality factors of transmon qubits. We subsequently verify our predictions by comparing them with measured transmon coherence. Transmon qubits of a particular design (Fig. 3a) were co-fabricated with the tripole striplines to ensure that the transmon validation devices were subjected to the same processing as the loss characterization devices. Tantalum-based transmons were fabricated by subtractively patterning the capacitor pads using tantalum, and Dolan bridge-style Al/AlO$_x$/Al Josephson junctions were additively patterned using double-angle shadow evaporation followed by liftoff (see Methods "Device fabrication"). Aluminum-based transmons were fabricated in the same way as the junctions in the tantalum-based process, except the capacitor and the Josephson junction were patterned in a single electron-beam lithography step.

To describe the losses in aluminum and tantalum-based transmon qubits, we once again invoke the participation ratio model (see Supplementary Table S8). The aluminum-based transmon is limited by the same sources of loss as aluminum tripole striplines: surface losses associated with the aluminum thin-film growth and patterning, bulk dielectric losses associated with the substrate, and package losses. The tantalum-based transmon has both tantalum and aluminum regions and is susceptible to surface loss associated with both the tantalum capacitor pads and the aluminum region near the junction.

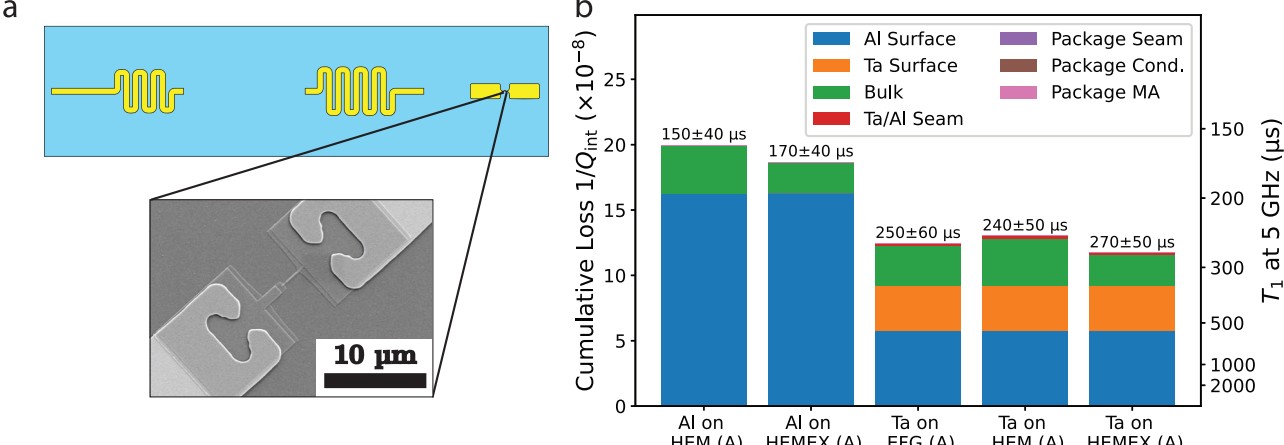

**Fig. 3 | Prediction of transmon loss. a** 3D transmon qubit design, from which the participation ratios were calculated. Inset: SEM of Josephson junction and near-junction region on a tantalum-based transmon. Ta leads to connect to the Al junction through an overlapping Ta/Al contact region. **b** Predicted loss and expected $T_1$ for transmons made using different materials and processes (Al vs. Ta capacitor pads). The loss budget is also computed, showing the dominant sources of loss in Al- and Ta-based transmons.

Additionally, due to the use of two separate metals deposited in different deposition steps, the contact between the tantalum and aluminum may also manifest loss in analogy with seam loss. Tantalum oxide or other contaminants located in the Ta/Al interface may contribute to an effective resistance in series with the Josephson junction. We characterized this loss in the microwave regime using a segmented stripline that is highly sensitive to Ta/Al contact loss and extracted a seam resistance of $260 \pm 47$ nΩ (see Supplementary Note 2: "Extracting Ta/Al contact loss"), which would limit the quality factor of the transmon to over $5 \times 10^8$, indicating that Ta/Al contact loss is negligible.

By combining the transmon participations with the extracted loss factors from the tripole and segmented striplines, we can compute the expected device coherence of aluminum- and tantalum-based transmon qubits on different types of annealed sapphire (Fig. 3b). Aluminum-based transmons are expected to achieve $T_1$ of 150–170 μs at 5 GHz, limited primarily by surface loss due to the aluminum-based process. By replacing the capacitor pads with tantalum using its respective process, the reduced surface loss is expected to yield dramatically improved $T_1$'s that exceed 240 μs, regardless of sapphire grade. However, nearly half of the tantalum-based transmon's loss is from the near-junction aluminum region, which is now the dominant factor that limits transmon relaxation. We attribute this to the small capacitance of the junction electrodes, which induces large electric fields that are localized in a small area, leading to high surface participation in the aluminum region (see Supplementary Table S8). Additionally, as new materials systems are developed that result in reduced surface loss, bulk dielectric loss begins to play a significant role. Already, bulk loss accounts for 15–20% of the tantalum-based transmon's loss; as a result, the microwave quality of the substrate must be considered as coherence continues to improve[49]. Finally, losses associated with the Ta/Al contact region and the package are predicted to be negligible; the first being due to the low Ta/Al contact resistance, and the second being due to the compact electromagnetic field profile of the transmon.

To verify the predicted transmon losses and validate our understanding of decoherence mechanisms and their roles in determining coherence, several aluminum- and tantalum-based transmons were fabricated on different grades of annealed sapphire, and their measured quality factors were compared with the ranges predicted using the transmon loss model. Consistent with the predicted transmon loss, representative $T_1$ measurements show an almost factor of two improvement in a tantalum-based transmon over an aluminum-based transmon (Fig. 4a). Each transmon's coherence was also measured over a period of at least 10 h to capture temporal fluctuations, and the predictive loss model showed remarkable consistency with the 90th percentile of transmon $T_1$, with the vast majority of measured $Q_{int}$ falling inside one standard deviation of the predicted $Q_{int}$ (Fig. 4b). These measured $Q_{int}$'s are also similar to those measured in other studies[36,37]. The choice of comparing 90th percentile $T_1$ measurements with the loss predictions was done to discount the effects of fluctuations of TLSs interacting in the region of the Josephson junction. Despite the statistical expectation of zero TLSs present in the junction[24,64–66], this region's small area and high energy density renders the transmon highly sensitive to deviations from that expectation due to stochastically fluctuating TLSs both in space and frequency[67,68] over long periods of time. As a result, the transmon $T_1$ can fluctuate tremendously over hours-long timescales (see Supplementary Note 5: "Temporal fluctuations of coherence in transmons, quantum memories, and resonators"). In contrast, this behavior is not seen in our resonators; resonator $Q_{int}$'s measured over long timescales fluctuate by ~±10%. We attribute this to the resonator's much larger area and more uniformly distributed electric field; single TLS fluctuations are not expected to dramatically affect resonator $Q_{int}$. As a result, loss factors extracted from resonator measurements can be used to predict the upper (90th percentile) range of $T_1$'s achievable by a transmon as its coherence fluctuates over long timescales.

## Optimized geometry to maximize coherence in a quantum memory

Our loss analysis has thus far shown that tantalum-based transmons can achieve high $T_1$'s but are significantly limited by surface participation near the junction. This motivates a more optimized design choice where we use a linear resonator to encode quantum information[41]. Linear resonators tend to have their electromagnetic fields distributed over a larger area, which leads to reduced surface participation and, therefore, a higher $Q_{int}$, regardless of what materials or fabrication processes are employed. Furthermore, the lack of a Josephson junction renders the resonator much less sensitive to TLS fluctuations, leading to more temporally stable coherence and dramatically suppressed pure dephasing. This has been demonstrated to great success using 3D cavity resonators as quantum memories[41–43], where logical qubits are encoded using the bosonic states of the resonator. While thin-film resonators have also been shown to be a viable candidate to be used as quantum

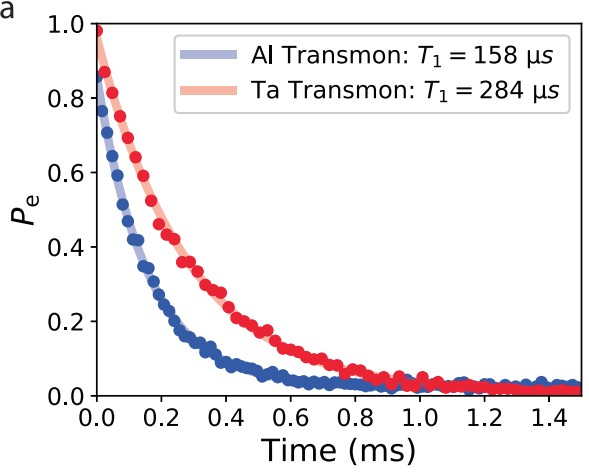
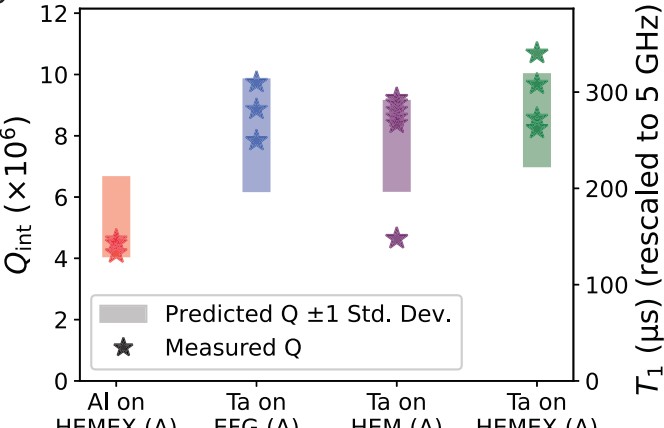

**Fig. 4 | Predicted vs. measured transmon quality factors. a** Representative Al- and Ta-based transmon $T_1$ curves showing an almost factor of 2 improvement by adopting a tantalum-based process. **b** Measured transmon $Q_{int}$ compared with predictions. Stars represent the 90th percentile transmon $Q_{int}$ of a distribution formed from repeated coherence measurements over a 10-h period. Shaded regions represent a predicted range spanning one standard deviation away from predicted transmon $Q_{int}$. Measured qubit frequencies ranged from 4.5 to 6.7 GHz.

memories[20,54], their coherence has thus far been far below their 3D counterparts. However, with the advancements in materials and fabrication processes demonstrated in this work, thin-film resonator $Q_{int}$'s exceeding $3 \times 10^7$ have been achieved at single-photon powers (Fig. 2a). It is, therefore, possible to optimize the design of a resonator to support a highly coherent on-chip quantum memory within the coaxial architecture. Such a device would have all the advantages of a planar device due to its more compact design and ability to be patterned with lithographic precision.

To implement a highly coherent on-chip quantum memory, we have developed the hairpin stripline, a multimode device whose fundamental mode is optimized to balance package and surface loss to maximize its $Q_{int}$ (Fig. 5a). The electromagnetic fields of this memory mode are localized primarily between the two arms of the hairpin, rendering it insensitive to package losses, while the large spacing between the two arms dilutes the electric field at the surfaces, thereby reducing surface loss. An ancilla transmon couples dispersively both to the memory mode to enable its fast control, and to the second-order mode of the hairpin stripline, which acts as a readout mode without needing to introduce additional hardware complexity (see Supplementary Table S11). The ancilla's capacitor pads are staggered with respect to each other in order to reorient its dipole moment to achieve the desired couplings to the electric fields of the hairpin modes (see

Supplementary Note 4: "Hairpin stripline device design and measurement").

To demonstrate the improvements in coherence achievable by optimizing materials and process choices, we apply the predictive loss model to the hairpin stripline and show that an aluminum-based process employed on unannealed HEM sapphire is not expected to produce remarkable coherence (Fig. 5c), and replacing the aluminum with a tantalum-based process leads to a modest expected improvement. Additional modest improvements are expected when annealed sapphire substrates are used; however, when both high-temperature substrate annealing and tantalum processes are employed, the hairpin stripline is expected to achieve a $T_1$ of $(1.1 \pm 0.2)$ ms, which rivals the coherence of commonly used quantum memories realized in 3D coaxial $\lambda/4$ post-cavities[41]. This dramatic improvement is only achieved when both materials and geometry are optimized to minimize both surface and package participation, resulting in a predominantly bulk loss-limited device.

Four hairpin stripline-based quantum memories were fabricated using a tantalum process on annealed HEMEX-grade sapphire substrates. The devices were measured in the same cylindrical tunnel packages used to measure the tripole striplines and transmon qubits. Memory $T_1$ and $T_2$ in the Fock ($|0\rangle,|1\rangle$) manifold were measured using the same pulse sequences as in ref. 41 (see Supplementary Fig. S8). Quantum memory coherence was remarkably consistent with predictions (Fig. 6); Fock state decay times were measured to be 1–1.4 ms. Additionally, measured Fock $T_2$ times approached $2T_1$, which bounds $T_\phi > 24$ ms, similar to 3D cavity-based quantum memories[43,69]. Additionally, continuous coherence measurements over 20 h showed minimal temporal fluctuations in $T_1$ and $T_2$; coherence fluctuated by no more than $\pm10\%$ over hours-long timescales, markedly different behavior from transmon qubits and consistent with a much-reduced sensitivity to TLS fluctuations (Supplementary Fig. S10c).

## Discussion

We have introduced a technique for characterizing microwave losses in thin-film resonators. We have shown that depending on resonator geometry, the surface, bulk, and package losses can be significant contributors to the total internal loss of a microwave resonator. We have also observed that our tantalum-based fabrication processes tend to yield higher internal quality factors due to improvements in surface quality and that annealing sapphire substrates results in dramatically reduced bulk dielectric loss tangents. Additionally, we have shown that by understanding sources of loss in resonators, we can make and experimentally verify predictions of losses in co-fabricated transmon qubits. By analyzing the sources of loss that limit state-of-the-art devices, we have utilized a powerful tool that revealed comprehensively what limits transmon coherence, and motivated the design of an optimized stripline-based quantum memory using thin-film superconductors patterned on a substrate. While our loss characterization results are specific to our materials and fabrication processes, the participation ratio model provides a versatile approach to loss characterization that can be adapted for the co-planar waveguide, flip-chip, or cavity-based cQED architectures; additional materials and loss channels can be straightforwardly studied by designing the appropriate participation matrix and introducing new devices or modes to characterize them (see Supplementary Note 2: "Extracting Ta/Al contact loss").

The implementation of a quantum memory in a stripline enables a coaxial architecture that is more scalable, more modular, and more compact than the more traditional cavity approach[20]. Ancilla-memory couplings can be lithographically defined, enabling far greater precision in device design. By employing a well-controlled fabrication process, consistently high device coherence can be achieved. Multiqubit systems can be more straightforwardly and compactly designed, as the striplines themselves are more compact than their 3D

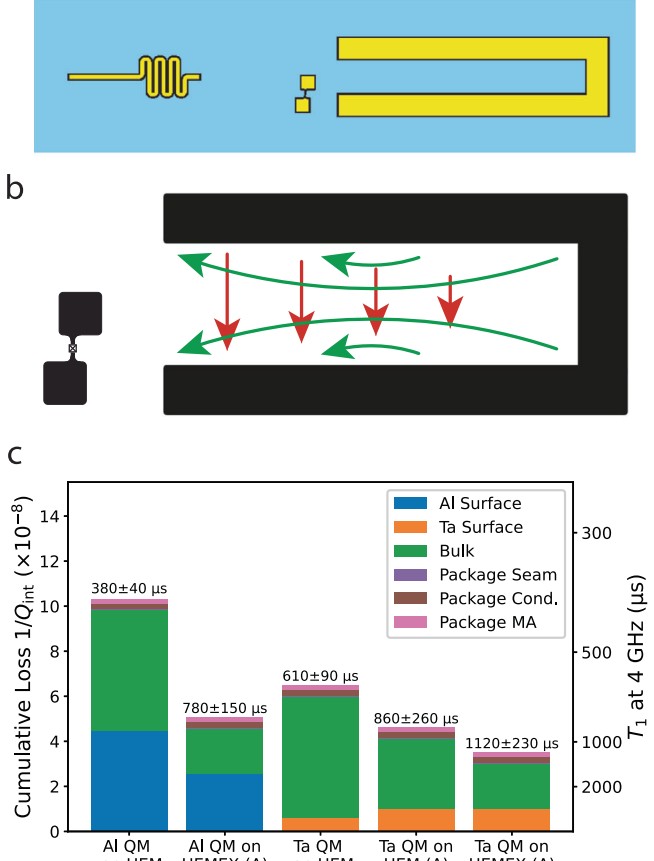

**Fig. 5 | Hairpin stripline quantum memory. a** Hairpin stripline quantum memory design. The ancilla transmon couples to the fundamental mode that acts as a storage resonator, and to the higher-order mode that acts as a readout resonator. A Purcell filter (meandered stripline on the left side of the chip) is used to enhance the external coupling of the readout mode. **b** Electric field behaviors of the memory mode (red arrows) and readout mode (green arrows). The ancilla's capacitor pads are staggered with respect to each other to adequately couple to both modes. **c** Predicted loss and expected T1 for hairpin striplines made using different substrate preparations and different superconducting thin films.

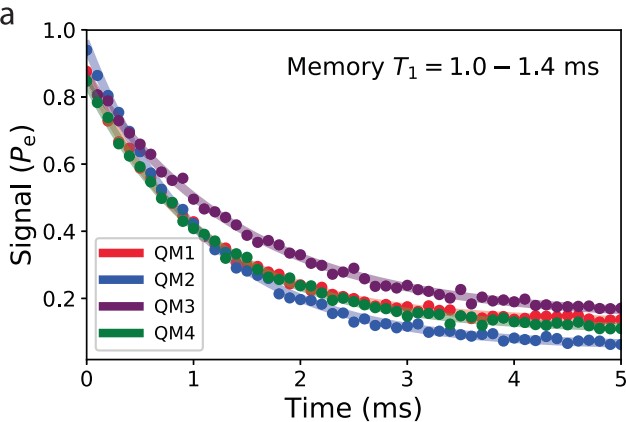

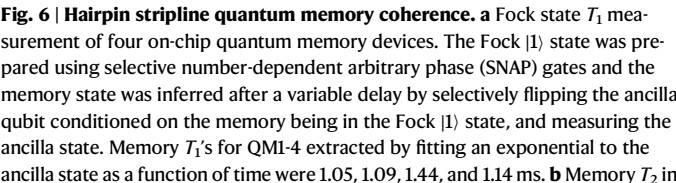

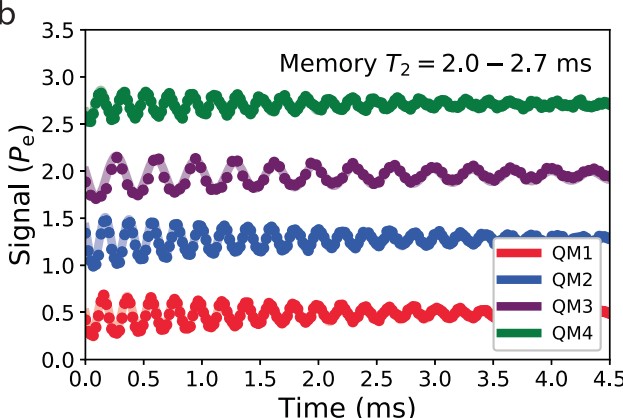

**Fig. 6 | Hairpin stripline quantum memory coherence. a** Fock state $T_1$ measurement of four on-chip quantum memory devices. The Fock $|1\rangle$ state was prepared using selective number-dependent arbitrary phase (SNAP) gates and the memory state was inferred after a variable delay by selectively flipping the ancilla qubit conditioned on the memory being in the Fock $|1\rangle$ state, and measuring the ancilla state. Memory $T_1$'s for QM1-4 extracted by fitting an exponential to the ancilla state as a function of time were 1.05, 1.09, 1.44, and 1.14 ms. **b** Memory $T_2$ in the Fock ($|0\rangle$,$|1\rangle$) manifold for the four devices measured in (**a**). The Fock state $\frac{1}{\sqrt{2}}(|0\rangle + |1\rangle)$ was prepared using SNAP gates and after a variable delay a small displacement was applied to interfere with the memory state, followed by measurement in the same way as in (**a**)[41]. Ancilla state as a function of time for QM2-4 were offset vertically by 0.75, 1.5, and 2.25, respectively, for visibility, and were fit to an exponentially decaying sinusoid. Extracted memory $T_2$'s for QM1-4 were 2.02, 2.00, 2.68, and 2.14 ms.

counterparts. Multiple devices can be fabricated on a single wafer and easily redesigned without modifying the package, allowing increased modularity. Additionally, the low pure dephasing observed in these devices allows for the implementation of noise-biased qubits, which can enable lower error correction thresholds toward the implementation of surface codes of dual-rail qubits[9,70]. Stripline-based quantum memories, therefore, provide a promising building block for realizing large-scale quantum computing with bosonic modes.

Finally, the loss characterization studies presented in this work have shown clear paths forward for improving coherence in superconducting qubits. Transmons are significantly limited by surface participation near the Josephson junction; as a result, developing better processes or using intrinsically lower-loss materials in this region may be critical toward improving transmon coherence to one millisecond and beyond. Additionally, improvements in surface loss must also accompany improvements in bulk dielectric loss; this is especially important for stripline-based quantum memories, which are dominated by bulk loss. This work demonstrates important techniques that help to provide an understanding of coherence-limiting mechanisms and inform optimization and design choices for superconducting quantum circuits.

## Methods
### Device fabrication
All devices were fabricated on c-plane sapphire substrates grown using either the edge-fed film growth (EFG) method or the heat-exchange method (HEM). HEMEX wafers were additionally graded HEM wafers based on superior optical properties[49,71]. All substrates were initially cleaned in a piranha solution (2:1 $H_2SO_4$:$H_2O_2$) for 20 min[36], followed by a thorough rinse in DI water for 20 min. Substrates were then optionally annealed in a FirstNano EasyTube 6000 furnace at 1200 °C in an oxygen-rich environment. The furnace was preheated to 200 °C and purged with nitrogen prior to wafer loading. The furnace was then purged with pure oxygen, followed by a gradual heating to 1200 °C at a controlled ramp of 400 °C/hr while continuously flowing oxygen. Once the furnace reached 1200 °C, the gas flows were shut off, and the wafers were allowed to anneal for 1 h in the oxygen-rich ambient conditions. Finally, the wafers were passively cooled over approximately 6 h by turning off the furnace heaters and flowing a 4:1 mixture of $N_2$:$O_2$ gas.

For tantalum-based devices, tantalum was deposited after the cleaning and optional annealing by DC magnetron sputtering while maintaining a substrate temperature of 800 °C. About 150 nm of tantalum was sputtered using an Ar pressure of 6 mTorr and a deposition rate of 2.5 Å/s. After deposition, the substrate was cooled at a controlled rate of 10 °C/min to prevent substrate damage due to the differential contraction of the Ta film and the sapphire surface. Tantalum films deposited this way were consistently in the (110) or (111)-oriented α-phase as shown by X-ray diffractometry (XRD) (Supplementary Fig. S12b) and have $T_c > 4.1$ K with RRR >15 (our best film has a $T_c = 4.3$ K and RRR = 55.8, see Supplementary Fig. S12a). To pattern the tantalum, the S1827 photoresist was spun on the wafer after deposition and patterned using a glass photomask and a Suss MJB4 contact aligner. After developing in Microposit MF319 developer for 1 min, the wafer was hard-baked for 1 min at 120 °C and treated with oxygen plasma using an AutoGlow 200 at 150 W and 300 mTorr $O_2$ for 2 min to remove resist residue. The tantalum was etched at a rate of 100 nm/min in an Oxford 80+ Reactive Ion Etcher using $SF_6$ with a flow rate of 20 sccm, a pressure of 10 mTorr, and an RF power of 50 W. After etching, the photoresist was removed by sonicating for 2 min each in $N$-methylpyrrolidone (NMP), acetone, isopropanol, and DI water. To remove any remaining organic residue, an additional 20 min piranha cleaning step was performed, and to remove excess tantalum oxide that may have grown due to the strong oxidizing nature of the piranha solution, an oxide strip was performed by dipping the wafer in Transene 10:1 BOE for 20 min[38,72], followed by a 20 min rinse in DI water.

Josephson junctions and aluminum devices were patterned using electron-beam lithography. The wafer was first dehydrated by baking at 180 °C for 5 min. Then, a bilayer of 700 nm MMA (8.5) MAA EL13 and 200 nm of 950K PMMA A4 was spun, with a 5 min bake at 180 °C following the spinning of each layer. To eliminate charging effects during electron-beam writing, a 15 nm aluminum anticharging layer was deposited by electron-beam evaporation. Electron-beam lithography was then performed using a Raith EBPG 5200+ to define the Dolan-bridge shadowmask. The anticharging layer was then removed by immersing the wafer in Microposit MF312 developer for 80 s, and the pattern was developed in 3:1 IPA:$H_2O$ at 6 °C for 2 min. The wafer was then loaded into the load-lock of a Plassys UMS300 electron-beam evaporator, where an Ar ion beam clean was performed at 400 V to remove the tantalum oxide and other surface residues prior to

aluminum deposition. The wafer was tilted by ±45 degrees and the ion beam cleaning was performed for 34 s at each angle in order to remove the oxide on the tantalum sidewall and to clean the region underneath the Dolan bridge. The same cleaning process was employed prior to the deposition of the aluminum-based devices. Following the ion beam clean, the wafer was transferred to the evaporation chamber where double-angle evaporation of aluminum was performed at ±25 degrees (20 nm followed by 30 nm) with an interleaved static oxidation step using an 85:15 $Ar:O_2$ mixture at 30 Torr for 10 min. After the second aluminum deposition, a second static oxidation step was performed using the same $Ar:O_2$ mixture at 100 Torr for 5 min in order to cap the surface of the bare aluminum with pure aluminum oxide. Liftoff was then performed by immersing the wafer in NMP at 90 °C for 1 h, followed by sonication for 2 min each in NMP, acetone, isopropanol, and DI water. The wafer was then coated with a protective resist before dicing into individual chips with in an ADT ProVectus 7100 dicer, after which the chips were cleaned by sonicating in NMP, acetone, isopropanol, and DI water.

## Device packaging

All striplines, transmons, and quantum memories were measured in cylindrical tunnel packages made out of conventionally machined high-purity (5N5) aluminum (Supplementary Fig. S2). The packages underwent a chemical etching treatment using a mixture of phosphoric and nitric acid (Transene Aluminum Etchant Type A) heated to 50 °C for 2 h[30]. The tunnels were ~34 mm long and 5 mm in diameter. Coupling was accomplished by a transverse feedline, allowing for multiple tunnels to be arranged side-by-side and measured in a multiplexed hanger configuration[20]; the same feedline is used for qubit, storage mode, and readout drives. The 40 mm × 4 mm chips on which the devices are fabricated are inserted into the tunnel package and clamped on either end by beryllium-copper leaf-springs. The clamps on either end of the tunnel also serve as end-caps for the tunnels themselves, thereby defining the locations of the seams and completing the enclosure.

## Measurement setup

A fridge wiring diagram can be found in Supplementary Fig. S1. Device packages are mounted to the mixing chamber stage of a dilution refrigerator operating at 20 mK. The packages are enclosed in multiple layers of shielding. First, a light-tight gold-plated copper shield internally coated with Berkeley black acts as an IR photon absorber[73]. A superconducting shield made of 1/64" thick lead foil is wrapped around the copper shield. Finally, a Cryoperm can serves as the outermost shield to attenuate the ambient magnetic fields at the package. Input lines are attenuated at both the 4 K stage (20 dB) and mixing chamber stage (50–60 dB depending on the line; 20 dB of reflective attenuation is achieved through the use of a directional coupler) and are filtered at multiple locations using 12 GHz K&L low-pass filters and custom-made eccosorb CR-110 IR filters. Output lines are also low-pass filtered and isolated from the devices using circulators and isolators. A SNAIL parametric amplifier (SPA) is used on the qubit output line to provide quantum-limited amplification for qubit readout. HEMT amplifiers at the 4 K stage provide additional low-noise amplification for the output signals.

Resonators are measured in the frequency domain using a vector network analyzer (Agilent E5071C). Qubits and quantum memories are measured in the time domain using an FPGA-based quantum controller (Innovative Integration X6-1000M), which can output arbitrary waveforms in pairs of I and Q quadratures at ≈50 MHz that are then upconverted to GHz frequencies using an LO tone generated by an Agilent N5183A (Readout drive uses a Vaunix LMS-103 for the LO) and a Marki IQ-0307-LXP mixer. Qubit, readout, and storage mode drives are all generated the same way and are combined and amplified using a Mini-Circuits ZVA-183-S+. The signals are finally attenuated by a room-temperature 3 dB attenuator to reduce the thermal noise temperature

before being fed into the fridge. Readout responses from the fridge are amplified with a room-temperature amplifier (MITEQ LNA-40-04001200-15-10P) and isolated before being downconverted using a Marki IR-0618-LXP mixer (the same LO is used for both the upconversion and downconversion of the readout signals). Downconverted signals are then amplified using a Mini-Circuits ZFL 500 before being fed into the ADC of the FPGA. All signal generator sources and VNA are clocked to a 10 MHz Rb frequency standard (SRS FS725).

## Calculation of participation ratios

Energy participation in various lossy regions are calculated using the commercial finite-element electromagnetic solver Ansys HFSS and the two-step meshing method detailed in ref. [19]. Thin-film conductors are approximated in a 3D electromagnetic simulation as perfectly conducting 2D sheets. Field behavior at the edges of the thin films are approximated using a heavily meshed 2D cross-sectional electrostatic simulation with explicitly defined surface dielectric regions of assumed thickness $t_{surf} = 3$ nm and relative permittivity $\epsilon_r = 10$ to maintain consistency with other works[38,44,45]. The true thickness and relative permittivity of these regions are unknown; while nanometer-scale microscopy of these interfaces can yield qualitative information about these interfaces, it cannot definitively reveal the dielectric properties or the presence or absence of physical signatures of loss. We, therefore, treat the true surface region thickness and relative permittivity as material/process parameters that re-scale the surface loss tangents and thereby define the intrinsic loss factor that corresponds to $p_{surf}$ as $\Gamma_{surf} = \sum_{k = SA,MS,MA} \frac{p_k}{p_{surf}} \frac{t_{k_0}}{t_{surf}} \frac{\epsilon_{r_0}}{\epsilon_r} \tan\delta_k$, where $\tan\delta_k$, $t_{k_0}$, and $\epsilon_{r_0}$ are the true dielectric loss tangent, thickness of the surface regions, and true dielectric constant, respectively[33,45].

We define a combined surface participation term, $p_{surf} = p_{SA} + p_{MS} + p_{MA}$ and define the corresponding surface loss factor as a weighted sum of the SA, MS, and MA loss factors (Supplementary Fig. S3). This construction of surface participation prevents us from distinguishing between the different surface losses, but because the relative scaling of these participations is roughly the same for all resonator geometries in this architecture, the geometric ratio $p_k/p_{surf}$ is geometry-independent; therefore, $\Gamma_{surf}$ still carries predictive power to estimate the loss of a desired resonator geometry. This formulation could also be modified to consider conductor loss in the thin films, whose participation scales similarly to the surface dielectric participations. In such a case, $\Gamma_{surf}$ is a surface loss factor that contains contributions from dielectric and conductor loss. Here, we assume conductor loss to be negligible, as aluminum thin films have been shown to have residual quasiparticle fractions as low as $x_{qp} = 5.6 \times 10^{-10}$ [73], where $x_{qp} \sim \Gamma_{cond}$. Assuming our tantalum films also have similarly low $x_{qp}$, we estimate the thin-film conductor loss to limit the tripole stripline modes to $Q_{int} > 10^{10}$.

We use the following integral equations to calculate the various on-chip and package participations in the coaxial tunnel architecture:

$$p_{SA,MS} = \frac{t_{surf} \int_{SA,MS} \epsilon_r \epsilon_0 |\overrightarrow{E}|^2 d\sigma}{\int_{all} \epsilon |\overrightarrow{E}|^2 dv} \tag{3}$$

$$p_{MA}, p_{pkg_{MA}} = \frac{t_{surf} \int_{MA} \epsilon_0 |\overrightarrow{E}_{vac}|^2 d\sigma}{\epsilon_{r,MA} \int_{all} \epsilon |\overrightarrow{E}|^2 dv} \tag{4}$$

$$p_{bulk} = \frac{\int_{bulk} \epsilon |\overrightarrow{E}|^2 dv}{\int_{all} \epsilon |\overrightarrow{E}|^2 dv} \tag{5}$$

$$p_{\text{pkg}_{\text{cond}}} = \frac{\lambda \int_{\text{surf}} \mu_0 |\vec{H}_{\parallel}|^2 d\sigma}{\int_{\text{all}} \mu_0 |\vec{H}|^2 dv} \qquad (6)$$

$$y_{\text{seam}} = \frac{\int_{\text{seam}} |\vec{J}_{\text{S}} \times \hat{l}|^2 dl}{\omega \int_{\text{all}} \mu_0 |\vec{H}|^2 dv}. \qquad (7)$$

For $p_{\text{SA,MS}}$, integration was done over a surface located 3 nm below the 2D sheet. For $p_{\text{MA}}$ and $p_{\text{pkg}_{\text{MA}}}$, integration was done over a surface located 3 nm above the 2D sheet. Because the MA surface dielectric region is not explicitly defined in the 3D simulation, the vacuum electric field was re-scaled to that of the MA field by invoking the continuity of the displacement field, $\epsilon_r \epsilon_0 E_{\text{MA},\perp} = \epsilon_0 E_{\text{vac},\perp}$. $p_{\text{pkg}_{\text{cond}}}$ was calculated by integrating the magnetic field energy density over the surface of the package wall and multiplying it by the effective penetration depth $\lambda$ of high-purity aluminum, which was previously measured to be $\approx 50$ nm[30]. Finally, seam loss is described using a seam admittance per unit length $y_{\text{seam}}$, which is a geometric factor analogous to a participation ratio, and a seam conductance per unit length $g_{\text{seam}}$, which is an intrinsic loss factor. $y_{\text{seam}}$ was calculated by integrating the current flow across the seam; both $y_{\text{seam}}$ and $g_{\text{seam}}$ have units $(\Omega m)^{-1}$ [50].

For transmons, a significant portion of the total magnetic energy is stored in the kinetic inductance of the Josephson junction; therefore, the total magnetic energy is calculated to include the energy stored in the junction, $U_{H_{\text{tot}}} = \int_{\text{all}} \mu_0 |\vec{H}|^2 dv + \frac{1}{2} L_J I_J^2$. Near-junction (<5-μm away) surface participations are calculated using an additional local 3D electrostatic simulation, and we invoke a similar argument as in ref. [19] and exclude the participation contribution from a region within 100 nm of the junction itself. This exclusion follows from the assumption that surface dielectric loss is dominated by a TLS density of -1 μm$^{-2}$GHz$^{-1}$, and therefore, the small region that is the junction itself should likely include zero TLSs and be lossless[24,64–66]. This assertion that the junction be lossless is further supported by earlier studies that have bounded the loss tangent of the junction oxide to below $4 \times 10^{-8}$ [74], and by recent quasiparticle tunneling experiments that have shown charge-parity switching lifetimes on the order of hundreds of milliseconds if the appropriate radiation shielding and microwave filtering are used[73], which has been replicated in this work (see Methods "Measurement setup").

### Extraction of loss factors using least-squares minimization

Starting with the matrix equation $\kappa_j = \sum_i P_{ji} \Gamma_i$, we use the least-squares fitting algorithm to extract the loss factors $\Gamma_i$ and propagate the measurement error $\sigma_{\kappa_j}$ onto the fit error $\sigma_{\Gamma_i}$[59]. If the rank of $\boldsymbol{P}$ is equal to or greater than the number of loss channels (i.e., $N_{\text{rows}} \geq N_{\text{columns}}$), the least-squares sum can be written down as:

$$S = \sum_j \left( \sum_i \tilde{P}_{ji} \Gamma_i - \tilde{\kappa}_j \right)^2, \qquad (8)$$

where $\tilde{P}_{ji} = P_{ji}/\sigma_{\kappa_j}$ and $\tilde{\kappa}_j = \kappa_j/\sigma_{\kappa_j}$ are the measurement-error-weighted participation matrix and internal loss, respectively. We can then express the least-squares sum in matrix form as $S = (\boldsymbol{\tilde{P}}\vec{\Gamma} - \vec{\tilde{\kappa}})^{\mathsf{T}} (\boldsymbol{\tilde{P}}\vec{\Gamma} - \vec{\tilde{\kappa}})$ and solve for $\vec{\Gamma}$ by setting $\partial S/\partial \vec{\Gamma} = 0$ to obtain $\vec{\Gamma} = \boldsymbol{C}\boldsymbol{\tilde{P}}^{\mathsf{T}} \vec{\tilde{\kappa}}$, where $\boldsymbol{C} = (\boldsymbol{\tilde{P}}^{\mathsf{T}} \boldsymbol{\tilde{P}})^{-1}$ is defined as the covariance matrix. We calculate the propagated error as $\vec{\sigma}^2_{\vec{\Gamma}} = \langle \delta\vec{\Gamma} \delta\vec{\Gamma}^{\mathsf{T}} | = \rangle \boldsymbol{C}\boldsymbol{\tilde{P}}^{\mathsf{T}} \langle \delta\vec{\tilde{\kappa}} \delta\vec{\tilde{\kappa}}^{\mathsf{T}} |\boldsymbol{\tilde{P}}\rangle \boldsymbol{C}^{\mathsf{T}}$, and $\langle \delta\tilde{\kappa}_i \delta\tilde{\kappa}_j | = \rangle \langle \frac{1}{\sigma_i} \frac{1}{\sigma_j} \delta\kappa_i \delta\kappa_j | = \rangle \delta_{ij}$, so $\vec{\sigma}^2_{\vec{\Gamma}} = \boldsymbol{C}\boldsymbol{\tilde{P}}^{\mathsf{T}} \boldsymbol{\tilde{P}} \boldsymbol{C}^{\mathsf{T}} = \boldsymbol{C}\boldsymbol{C}^{-1} \boldsymbol{C}^{\mathsf{T}} = \boldsymbol{C}$. Therefore, the propagated error on the extracted loss factors are given by $\sigma_{\Gamma_i} = \sqrt{C_{ii}}$.

### Subtraction of package conductor and dielectric losses

Package losses are comprised of conductor, surface dielectric (MA), and seam losses. To quantify the conductor and MA losses, we use previously obtained loss factors for conventionally machined 5N5 aluminum, measured using a multimode resonator made entirely of 5N5 aluminum called the forky whispering-gallery-mode resonator[33]. From the extracted losses of the two measured devices (F1(e) and F2(e)) we obtain $R_s = (0.61 \pm 0.28)$ μΩ and $\tan\delta_{\text{pkg}_{\text{MA}}} = (4.1 \pm 1.8) \times 10^{-2}$, where $\Gamma_{\text{pkg}_{\text{cond}}} = R_s/(\mu_0 \omega\lambda)$ and $\Gamma_{\text{pkg}_{\text{MA}}} = \tan\delta_{\text{pkg}_{\text{MA}}}$, where $R_s$ is the surface resistance of the superconductor, $\lambda$ is the effective penetration depth of the superconductor, $\omega$ is the frequency of the resonant mode, and $\tan\delta$ is the surface dielectric loss tangent of the MA interface. Applying these loss factors to the tripole striplines measured in Fig. 2a, we obtain a package loss limit due to conductor and MA dielectric loss to be $1/Q_{\text{D1}} = 1/17 \times 10^9$, $1/Q_{\text{D2}} = 1/4.7 \times 10^8$, and $1/Q_{\text{C}} = 1/1.3 \times 10^8$ for the D1, D2, and C modes, respectively. These package loss contributions indicate that they can be treated as residual losses, as they account for no more than 10–15% of the total loss of the common mode, with seam losses being the dominant source of package loss.

### Measurement of resonator quality factor

Microwave resonators were measured in the frequency domain using a vector network analyzer (VNA). The scattering parameter $S_{21}$ describes the response to driving the resonator as a function of frequency and is given by

$$S_{21}(\omega) = a e^{i\alpha} e^{-i\omega\tau} \left[ 1 - \frac{(Q_L/|Q_c|)e^{i\phi}}{1 + i2Q_L(\omega/\omega_r - 1)} \right], \qquad (9)$$

where $a$ is the total attenuation of the line, $\alpha$ is the global spurious phase shift, $\tau$ is the electrical delay, $\omega_r$ is the resonance frequency, and $Q_c = |Q_c| e^{-i\phi}$ is a complex coupling quality factor where $\phi$ describes the asymmetry in the hanger response[56]. The real-valued loaded quality factor $Q_L$ is the total quality factor due to both internal and external (coupling) loss, $1/Q_L = 1/Q_{\text{int}} + \cos\phi/|Q_c|$, where $Q_{\text{int}}$ is the internal quality factor due to intrinsic material and process-based losses. The fitting methods used in ref. [56] are robust in that fitting resonators that are overcoupled or undercoupled by as much as a factor of 10 is readily possible. Resonators measured in this work had $|Q_c| = 2 - 10 \times 10^6$ and were, therefore, never too overcoupled or undercoupled. The excitation field of the resonator is determined by the input power $P_{\text{in}}$ and can be expressed in terms of an average photon number in the resonator as $\bar{n} = \frac{2}{\hbar\omega_r^2} \frac{Q_L^2}{Q_c} P_{\text{in}}$ (see Supplementary Note 10: "Derivation of resonator average photon number").

### Transmon coupling quality factor

Measured transmon $T_1$ is proportional to the loaded quality factor of the mode, $(\omega T_1)^{-1} = Q_L^{-1} = Q_{\text{int}}^{-1} + Q_c^{-1}$. The quality factor predictions made in Fig. 3b are based on internal losses only; therefore, the coupling quality factor must be measured for transmons in order to properly compare predicted $Q_{\text{int}}$ with measured $Q_{\text{int}}$. While this can be done using a finite-element electromagnetics solver, the true $Q_c$ is dependent on the transmon chip's placement within the tunnel package and can vary by as much as 50% if the chip's position varies by as little as 0.5 mm from the nominal. We, therefore, determined the $Q_c$ in situ by calibrating the qubit Rabi rate in the $g$-$e$ manifold as a function of drive power. The bare transmon Hamiltonian in the presence of a drive can be expressed as

$$H = \hbar\omega_q \hat{a}^\dagger \hat{a} - E_J \left[ \cos\hat{\Phi}_q + \left( 1 - \frac{1}{2}\hat{\Phi}_q^2 \right) \right] + \hbar\Omega_{\text{Rabi}} \cos\omega_d t \left( \hat{a}^\dagger + \hat{a} \right), \qquad (10)$$

where $\hat{a}$, $\omega_q$, $E_J$, and $\hat{\Phi}_q$ represent the transmon ladder operator, qubit transition frequency, Josephson energy, and flux operator, respectively. The term in the square brackets describes the nonlinearity of the transmon, and is assumed to be small enough such that it can be applied perturbatively towards a simple harmonic oscillator Hamiltonian. The drive can be parameterized by a drive strength or Rabi rate $\Omega_{\text{Rabi}}$ and a drive frequency $\omega_d$. We move into the rotating frame of the drive, followed by the rotating frame of the transmon and the displaced frame of the drive to arrive at the following transformed Hamiltonian $\tilde{H}$:

$$\tilde{H} = -E_J \cos\hat{\tilde{\Phi}}_q - E_J\left(1 - \frac{1}{2}\hat{\tilde{\Phi}}_q^2\right), \tag{11}$$

where $\hat{\tilde{\Phi}}_q = \phi_q(\tilde{a}^\dagger + \tilde{a} - \xi^* - \xi)$, and $\xi(t) = -\frac{i\Omega_{\text{Rabi}}e^{-i\omega_d t}}{\omega/Q_L + i2\Delta}$, where $\Delta = \omega_q - \omega_d$ and $|\xi|^2$ is the photon number. Since the transmons are driven in a hanger configuration and can be approximated as a harmonic oscillator as long as leakage to higher computational states is negligible, we can relate the photon number to $Q_c$ by $\overline{n} = \frac{2}{\hbar\omega_r^2}\frac{Q_L^2}{Q_c}P_{\text{in}}$. We can therefore derive the relation between the coupling Q and the qubit Rabi rate to be $Q_c = \frac{2P_{\text{in}}}{\hbar\Omega_{\text{Rabi}}^2}$. From this relation, we measure transmon $Q_c$ to vary between $30-70 \times 10^6$ due to variations in chip positioning within the tunnel, where the nominal positioning was simulated to yield $Q_c \approx 40 \times 10^6$. For our highest Q transmons, the external loss accounts for as much as 25% of the total loss. A solution to this extra loss is to simply undercouple the transmons even more from the drive line.

The $Q_c$ for the hairpin striplines, on the other hand, were simulated to be ~$10^9$. Imprecision in chip positioning can also lead to significant variations in $Q_c$ for this device; the resulting $Q_c$ can vary between $5-15 \times 10^8$. However, the hairpin striplines have measured $Q_L = 25-35 \times 10^6$; we, therefore, estimate that coupling loss accounts for less than 5% of the total loss of the hairpin stripline quantum memories.

## Data availability
The data presented in this study is available at https://doi.org/10.6084/m9.figshare.25426141 and more detailed source data is available from the corresponding authors upon request.

## Code availability
Codes are available upon request.

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

## Acknowledgements

We thank Andrew Houck, Nathalie de Leon, Alex Place, and Aveek Dutta for helpful discussions on the fabrication and surface processing of tantalum-based transmons. We also thank Michael Hatridge for providing us with a working Ar ion-beam cleaning recipe. We are grateful for the assistance lent by our colleagues Aniket Maiti, John Garmon, and Jacob Curtis in troubleshooting our room-temperature electronics apparatus for qubit and quantum memory measurement. We thank Tom Connolly for useful discussions about radiation shielding and RF filtering. We thank Nico Zani for useful discussions on simulating and calculating energy participation ratios. Finally, we thank Yong Sun, Kelly Woods, Lauren McCabe, Michael Rooks, and Sean Reinhart for their assistance and guidance in developing and implementing device fabrication processes. This research was

supported by the US Army Research Office (ARO) under grants W911-NF-18-1-0212 and W911-NF-23-1-0051 and by the US Department of Energy, Office of Science, National Quantum Information Science Research Centers, Co-design Center for Quantum Advantage (C2QA) under contract No. DE-SC0012704. This research used the Electron Microscopy and Materials Synthesis & Characterization facilities of the Center for Functional Nanomaterials (CFN), which is a U.S. Department of Energy Office of Science User Facility, at Brookhaven National Laboratory under Contract No. DE-SC0012704. The views and conclusions contained in this document are those of the authors and should not be interpreted as representing official policies, either expressed or implied, of the ARO or the US Government. The US Government is authorized to reproduce and distribute reprints for Government purposes, notwithstanding any copyright notation herein. Fabrication facilities use was supported by the Yale Institute for Nanoscience and Quantum Engineering (YINQE) and the Yale SEAS Cleanroom.

## Author contributions

S.G. designed and simulated the tripole stripline, transmon qubits, and hairpin stirpline quantum memories and fabricated the Ta-based devices, while Y.L. fabricated the Al-based devices and implemented the measurement room-temperature electronics apparatus. S.G. measured the transmons and quantum memories, while Y.W. measured the tripole stripline resonators. S.G. and Y.W. performed loss analysis and loss budgeting. A.B. assisted with device fabrication and measurement. C.U.L and L.K assisted with device design and error analysis. K.K. performed TEM. C.Z., R.L., Y.J. and M.L. performed XRD and DC resistivity measurements. S.G., Y.W., L.F. and R.J.S. wrote the manuscript with feedback from all co-authors.

## Competing interests

L.F. and R.J.S. are founders and shareholders of Quantum Circuits Inc. (QCI). The remaining authors declare no competing interests.
