## [Peer Review File · Nature Communications]

REVIEWER COMMENTS

Reviewer #1 (Remarks to the Author):

Summary

This paper addresses the challenge of understanding and mitigating losses in superconducting circuits, a critical aspect for enhancing relaxation times. The study employs multimode resonators in a controlled environment, specifically a cylindrical tunnel with capacitive driving in a hangar configuration. These resonators exhibit tripole stripline geometry, comprising two differential modes and one common mode. Surface losses predominantly influence the internal quality factor of the first differential mode (D1), while both surface and bulk losses impact the second differential mode (D2). The electric field distribution of the common mode (C) causes the package seam loss to be the dominant factor determining internal quality factor. The concurrent evaluation of quality factors across these modes on a single chip allows for the estimation of both surface and bulk losses and other loss sources across different metal and substrate materials.

To validate the loss model derived from the tripole resonators, the authors applied this knowledge to optimize the coherence time of a transmon qubit and a hairpin stripline resonator. The most favorable qubit performance was achieved using a chip constructed from Tantalum on annealed HEMEX-grade sapphire, yielding a T_1 of 284 microseconds, in line with top-tier global benchmarks. The final design of the hairpin stripline resonator, also utilizing Tantalum on annealed sapphire, exhibited a relaxation time between 1 and 1.4 milliseconds, consistent with the theoretical estimate of 1.1 milliseconds. Additionally, the resonator demonstrated a T_2 time ranging from 2.0 to 2.7 milliseconds, positioning it as a promising candidate for quantum memory applications. This study's principal outcome underscores that optimized 2D resonators can rival the performance of 3D coaxial cavities, offering substantial benefits for future quantum processors based on bosonic systems.

Review

The paper is well-crafted, featuring results that are presented in a lucid and intuitive manner, complemented by elucidating diagrams. The supplementary section serves as a valuable resource, furnishing crucial details that enhance the comprehensibility of the main text, particularly for readers possessing technical expertise. Considering these merits, I am of the opinion that this manuscript aligns well with the standards of Nature Communications. I have a few minor comments that the authors should address.

- Line 22, left column: the authors should add a relevant reference arXiv: 2304.13257 demonstrating surface encapsulation as a method to improve coherence
- Line 218 left column: y_{seam} and g_{seam} are not defined.
- Fig. 5: I think an illustration of the electric field distribution for the hairpin resonator would be useful in the main text. Similarly, a single line explaining the staggering of the capacitor pads in the main text would be nice.
- The SNAP-displacement pulse sequences used for preparing Fock state $|1\rangle$ and $|0\rangle+|1\rangle$ in the supplement would be useful.

Reviewer #2 (Remarks to the Author):

In this study, three types of superconducting devices were prepared. The first type is the tripole stripline structure, which is a superconducting microwave resonator with multiple modes for differentiating between surface loss, bulk dielectric loss, and package loss in thin-film superconducting quantum circuits. By analyzing the resonant quality factors of its three modes with a pre-knowledge of the geometric energy participation ratio, values of loss tangents in various conditions can be estimated. The authors performed quantitative comparisons of surface loss, substrate loss, and packaging loss due to a number of factors including metal materials (tantalum vs. aluminum), substrate grade, substrate treatment, and packaging setup. Based on these comparisons, the authors designed and prepared a second type of devices that incorporate transmon qubits, and predicted qubit energy relaxation times using previously obtained loss tangents, which were compared with experimental results. In the third type of device, a quantum memory structure was employed to minimize both surface and package losses for a longer energy relaxation time. This structure featured a specifically-designed Hairpin stripline with tantalum on annealed HEMEX-grade sapphire, whose energy relaxation time exceeded 1 ms while pure dephasing was negligible as characterized experimentally.

Fighting decoherence induced by dielectric two-level states (TLS) either in bulk or on surface is a crucial subject for improving the ultimate performance of superconducting qubit devices. However, it is notoriously hard to tack this issue, since many hidden or uncontrollable variables are likely associated with the long sequence of fabrication and characterization processes, due to which device-to-device or run-to-run deviations could be large and unavoidable. It is extremely difficult (even meaningless to some degree) to try to accurately quantify losses and draw affirmative conclusions despite the vast literature available on this topic.

I do appreciate the huge amount of effort that the authors spent and some useful information obtained from their work. The novelty in methodology comes at the tripole stripline design, for which relevant TLS dissipation channels (surface, bulk, and package) can be modeled in a simplified version and roughly quantified within a single device. Other analyzing tools used in this work, such as calculation of the energy participation ratio, the empirical formula describing the power-dependent TLS loss, and the measurement of resonator quality factor, have been well-documented in literature as first appeared in a series of works mostly published in applied physics letters more than 10 years ago. I am a bit disappointed concerning the useful conclusions drawn in this work, as minimizing surface participation ratio with redesigns and replacing Al with Ta are two well-known advices in the community that could improve the device coherence. The additional take-home message gained from this work is related to the package losses in the coaxial architecture, which could be a potential source of large device-to-device variation in measuring the intrinsic loss tangent. The quantum memory structure with $T_1 > 1\text{ms}$ is a remarkable technical advancement, which builds upon findings already reported in the rich literature and those by the authors. This work with summary of loss tangents in various conditions also provides a valuable reference guide for the community.

However, I have a serious complaint on how the authors carried out their data analysis throughout the manuscript.

Figure 2a shows the power dependence of internal quality factor of the modes of a particular tripole stripline device. It is strange that error bars vary significantly crossing curves even for similar Q values. Usually, measurement at high powers should have less uncertainty for a single curve, which is not the case here. The authors have to clarify the coupling Q values obtained for these curves, and explain the error bars in more details.

In Fig. 2b, the authors claimed that "propagated errors are small ($\sim 10\%$) and are hidden within the width of the solid lines". I completely disagree on this type of statement, which could be misleading (it gives me the impression that bulk loss is power independent). Since data yielding these loss factors in Fig. 2a already show large uncertainties ($>20\%$) at low powers (D2 and C modes), plots in Fig. 2b should first convey this level of uncertainty. In addition, to extract loss factors, the authors used a participation ratio model assuming thin (approximately 3 nm) dielectric (relative permittivity ~ 10) regions and simplified the surface loss by a weighted sum (it might be over simplified since weighted sum could be different for different modes). Anyway, the modeling itself could introduce some uncertainties that the authors might have ignored. I would like to point out that knowing the existence of these errors is necessary since all subsequent calculations are based on the values listed in the supplementary tables.

In Fig. 2d, Ta films on unannealed substrates tend to exhibit smaller surface losses compared to those on annealed substrates. This is confusing! In Tab. S1, the surface loss and bulk loss data are exactly

the same for Ta on EFG-sapphire substrates with and without annealing. This mistake needs be corrected for me to understand Fig. 2d.

In Figs. 3b and 5b, how were the predictions on the lifetimes of transmon qubit and quantum memory made? According to the supplementary tables, values of the loss factors in the same category vary significantly from device to device. Calculations based on the average of few scattered samples are always not that trustable. Without fully understanding the errors of the listed values in the supplementary tables for the abovementioned reasons, I think the claim "With this knowledge we predict and experimentally verify the relaxation times of aluminum- and tantalum-based transmon qubits" should be softened in tone.

I am puzzled why the x-indices in Figs. 3b and 5b were listed differently.

Reviewer #1 (Remarks to the Author):

Reviewer's comments in black; author's comments in red

Summary:

This paper addresses the challenge of understanding and mitigating losses in superconducting circuits, a critical aspect for enhancing relaxation times. The study employs multimode resonators in a controlled environment, specifically a cylindrical tunnel with capacitive driving in a hangar configuration. These resonators exhibit tripole stripline geometry, comprising two differential modes and one common mode. Surface losses predominantly influence the internal quality factor of the first differential mode (D1), while both surface and bulk losses impact the second differential mode (D2). The electric field distribution of the common mode (C) causes the package seam loss to be the dominant factor determining internal quality factor. The concurrent evaluation of quality factors across these modes on a single chip allows for the estimation of both surface and bulk losses and other loss sources across different metal and substrate materials.

To validate the loss model derived from the tripole resonators, the authors applied this knowledge to optimize the coherence time of a transmon qubit and a hairpin stripline resonator. The most favorable qubit performance was achieved using a chip constructed from Tantalum on annealed HEMEX-grade sapphire, yielding a T1 of 284 microseconds, in line with top-tier global benchmarks. The final design of the hairpin stripline resonator, also utilizing Tantalum on annealed sapphire, exhibited a relaxation time between 1 and 1.4 milliseconds, consistent with the theoretical estimate of 1.1 milliseconds. Additionally, the resonator demonstrated a T2 time ranging from 2.0 to 2.7 milliseconds, positioning it as a promising candidate for quantum memory applications. This study's principal outcome underscores that optimized 2D resonators can rival the performance of 3D coaxial cavities, offering substantial benefits for future quantum processors based on bosonic systems.

Review:

The paper is well-crafted, featuring results that are presented in a lucid and intuitive manner, complemented by elucidating diagrams. The supplementary section serves as a valuable resource, furnishing crucial details that enhance the comprehensibility of the main text, particularly for readers possessing technical expertise. Considering these merits, I am of the opinion that this manuscript aligns well with the standards of Nature Communications. I have a few minor comments that the authors should address.

Author Reply: We thank the reviewer for the feedback, and are happy to hear that the manuscript was clear and informative. We will address each comment below.

1. Line 22, left column: the authors should add a relevant reference arXiv: 2304.13257 demonstrating surface encapsulation as a method to improve coherence

We have added this reference in line 28 of our revised manuscript.

2. Line 218 left column: y_{seam} and g_{seam} are not defined.

We understand that while y_{seam} and g_{seam} are not explicitly defined in the main text, they are quantities that have been well-defined in many other works, two of which we have cited (see refs. [33] and [50]); as a result, we felt that they did not need to be defined in the main text. However, we have added a reference to the Methods section “Calculation of participation ratios” in line 226 in order to orient the reader towards a more detailed description of seam loss. In this section, the calculation of y_{seam} is given explicitly by Eqn (7), and we have added a short description of y_{seam} and g_{seam} in lines 1017-1021, reproduced here:

Finally, seam loss is described using a seam admittance per unit length y_{seam} , which is a geometric factor analogous to a participation ratio, and a seam conductance per unit length g_{seam} , which is an intrinsic loss factor. y_{seam} was calculated by integrating the current flow across the seam; both y_{seam} and g_{seam} have units Ωm [50].

3. Fig. 5: I think an illustration of the electric field distribution for the hairpin resonator would be useful in the main text. Similarly, a single line explaining the staggering of the capacitor pads in the main text would be nice.

We have included an explanation of the staggered capacitor pads in lines 642-647, reproduced here:

The ancilla's capacitor pads are staggered with respect to each other in order to reorient its dipole moment to achieve the desired couplings to the electric fields of the hairpin modes (see Supplementary Note 4: “Hairpin stripline device design”).

We have also added an illustration of the electric field behavior of the hairpin stripline memory and readout modes in figure 5

4. The SNAP-displacement pulse sequences used for preparing Fock state $|1\rangle$ and $|0\rangle+|1\rangle$ in the supplement would be useful.

We have included Supplementary Figure S9, which shows the pulse sequences for the Fock T1 and T2 measurement.

Reviewer #2 (Remarks to the Author):

Reviewer's comments in black; author's comments in red

In this study, three types of superconducting devices were prepared. The first type is the tripole stripline structure, which is a superconducting microwave resonator with multiple modes for differentiating between surface loss, bulk dielectric loss, and package loss in thin-film superconducting quantum circuits. By analyzing the resonant quality factors of its three modes with a pre-knowledge of the geometric energy participation ratio, values of loss tangents in various conditions can be estimated. The authors performed quantitative comparisons of surface loss, substrate loss, and packaging loss due to a number of factors including metal materials (tantalum vs. aluminum), substrate grade, substrate treatment, and packaging setup. Based on these comparisons, the authors designed and prepared a second type of devices that incorporate transmon qubits, and predicted qubit energy relaxation times using previously obtained loss tangents, which were compared with experimental results. In the third type of device, a quantum memory structure was employed to minimize both surface and package losses for a longer energy relaxation time. This structure featured a specifically-designed Hairpin stripline with tantalum on annealed HEMEX-grade sapphire, whose energy relaxation time exceeded 1 ms while pure dephasing was negligible as characterized experimentally.

Fighting decoherence induced by dielectric two-level states (TLS) either in bulk or on surface is a crucial subject for improving the ultimate performance of superconducting qubit devices. However, it is notoriously hard to tack this issue, since many hidden or uncontrollable variables are likely associated with the long sequence of fabrication and characterization processes, due to which device-to-device or run-to-run deviations could be large and unavoidable. It is extremely difficult (even meaningless to some degree) to try to accurately quantify losses and draw affirmative conclusions despite the vast literature available on this topic.

I do appreciate the huge amount of effort that the authors spent and some useful information obtained from their work. The novelty in methodology comes at the tripole stripline design, for which relevant TLS dissipation channels (surface, bulk, and package) can be modeled in a simplified version and roughly quantified within a single device. Other analyzing tools used in this work, such as calculation of the energy participation ratio, the empirical formula describing the power-dependent TLS loss, and the measurement of resonator quality factor, have been well-documented in literature as first appeared in a series of works mostly published in applied physics letters more than 10 years ago. I am a bit disappointed concerning the useful conclusions drawn in this work, as minimizing surface participation ratio with redesigns and replacing Al with Ta are two well-known advices in the community that could improve the device coherence. The additional take-home message gained from this work is related to the package losses in the coaxial architecture, which could be a potential source of large device-to-device variation in measuring the intrinsic loss tangent. The quantum memory structure with $T_1 > 1\text{ms}$ is a remarkable technical advancement, which builds upon findings already reported in the rich literature and those by the authors. This work with summary of loss tangents in various conditions also provides a valuable reference guide for the community.

However, I have a serious complaint on how the authors carried out their data analysis throughout the manuscript.

Author Reply: We thank the reviewer for the feedback and well-informed review. We would first like to address a point the reviewer made in their review, mainly, “I am a bit disappointed concerning the useful conclusions drawn in this work, as minimizing surface participation ratio with redesigns and replacing Al with Ta are two well-known advices in the community that could improve the device coherence.” We would like to clarify here that the conclusion from our work is not necessarily to improve coherence by minimizing surface participation ratio. Our optimized device, the hairpin stripline, does not minimize surface participation. In fact, if we were to do so, we would have had reduced coherence. This is because designs that minimize surface participation tend to have increased sensitivity to package losses. The hairpin stripline achieved millisecond coherence times because we balanced the loss contributions from the surface, the bulk, and the package. The result was a device that had higher surface participation than what we could otherwise achieve, in order to reduce package loss. Additionally, the optimization of materials choices was not as simple as replacing Al with Ta. We also utilized annealed sapphire substrates in order to reduce bulk dielectric loss. As a result, the improvements in device coherence come not just from reducing surface losses and using Ta, but also from balancing loss contributions from the package and surface, and by utilizing lower-loss substrates. While this might seem like an obvious strategy to an expert in the field, we feel as though it is not well appreciated by the community, as many in the field focus primarily on reducing surface losses with less focus on other loss mechanisms. Here, we challenge this notion by showing systematically that other sources of loss matter, and by comprehensively characterizing them we can improve coherence in a predictable way. We therefore believe that this work and the conclusions drawn within are useful for the superconducting qubits and cQED community.

We will address each point of concern below.

1. Figure 2a shows the power dependence of internal quality factor of the modes of a particular tripole stripline device. It is strange that error bars vary significantly crossing curves even for similar Q values. Usually, measurement at high powers should have less uncertainty for a single curve, which is not the case here. The authors have to clarify the coupling Q values obtained for these curves, and explain the error bars in more details.

The error bars vary between curves because the coupling quality factor is different for each of the modes. The D2 and C modes in particular were in the overcoupled regime (coupling quality factors of 2.2M and 2.0M, respectively). As a result, the resonator circle fit propagates a larger error onto the extracted internal Q. For the D2 mode (14x overcoupled), this statistical effect dominates the error bars and explains why the error on internal Q is increased slightly. The C mode (8x overcoupled) is not as overcoupled as the D2 mode; therefore, the internal Qs at higher powers can be extracted with minimal propagated error. At lower powers, the error is dominated by low measurement SNR.

We have included the coupling quality factors for the measured device of figure 2a in the caption, reproduced here:

The coupling quality factors Q_c for this device are 6.3×10^6 , 2.2×10^6 , and 2.0×10^6 for the D1, D2, and C modes, respectively. The relatively large error bars on the measured Q_{int} of the D2 and C modes (fractional errors of 7% and 17%, respectively, at single photon powers) can be attributed to these modes being in the overcoupled regime and reduced signal-to-noise ratio at low excitation powers.

2. In Fig. 2b, the authors claimed that "propagated errors are small ($\sim 10\%$) and are hidden within the width of the solid lines". I completely disagree on this type of statement, which could be misleading (it gives me the impression that bulk loss is power independent). Since data yielding these loss factors in Fig. 2a already show large uncertainties ($>20\%$) at low powers (D2 and C modes), plots in Fig. 2b should first convey this level of uncertainty. In addition, to extract loss factors, the authors used a participation ratio model assuming thin (approximately 3 nm) dielectric (relative permittivity ~ 10) regions and simplified the surface loss by a weighted sum (it might be over simplified since weighted sum could be different for different modes). Anyway, the modeling itself could introduce some uncertainties that the authors might have ignored. I would like to point out that knowing the existence of these errors is necessary since all subsequent calculations are based on the values listed in the supplementary tables.

To clarify, our intention with this comment was simply to state that the propagated errors are difficult to distinguish beyond the width of the solid lines. The loss factors in fig. 2b are plotted on a log scale, and as a result even error bars of 20% would be difficult to distinguish here. We have also been very careful of our wording in lines 322-336 in order to not mislead the reader. We specifically avoid saying that bulk loss is completely power independent; rather, we say that the surface loss factor is heavily power dependent while the bulk loss factor is relatively much less so. This is consistent with the different degrees of power dependence in the D1, D2, and C modes, in that the mode with the highest surface participation has the most power dependence.

To address the reviewer's concern about the large uncertainties, we revisited the data and can confirm that the relative uncertainties of the single-photon quality factors of the D2 and C modes for the device shown in fig. 2a are 7% and 17%, respectively (see reply to first comment). To address the loss factor error propagation in fig. 2b, we want to refer the reviewer to lines 308-313. Here, we explain how we use the TLS model to fit the measurement data and use the extracted TLS fit parameters to interpolate the values of Q that are then used for the loss factor analysis. The TLS fit was also error-weighted, meaning that data points with large uncertainty are weighted less in the fitting algorithm. Additionally, because we are fitting only 4 parameters using 10-12 data points, we are able to constrain the TLS fit parameters with statistical uncertainty that is actually less than that of the measured data points. As a result, the propagated error on the interpolated Q values used for loss factor extraction can be less than the measurement uncertainty. This error ($\sim 10\%$) is conveyed in fig 2b and is barely visible on the log-scale plot of the loss factors; the exact values are listed in Table S1. We note that this is a purely statistical argument; however, this analysis is self-consistent in that the extracted loss

factors of figure 2b can accurately reproduce the power dependence of the three modes shown in figure 2a. This subsequently allows us to calculate the loss budget shown in fig. 2c. It should be emphasized here that the usefulness of these specific loss factors extracted from a single device is itself minimal, because it does not capture device-to-device variations due to uncontrolled variables in the device fabrication and packaging process. To do this, we measured multiple nominally identical devices to make capture the variation, which we have found to be greater than the statistical uncertainty of the extracted loss factors from individual devices. This is shown in figs 2d and 2e, and in Table S3, where the surface and bulk loss factor variations can range from 20% to 40%. These are the relevant uncertainties that we use to predict the range of expected Qs for transmons and hairpin stripline quantum memories.

To address the reviewer's concern about the simplification of surface loss employed in this work, we refer the reviewer to lines 968-988 in the methods section that addresses this directly, which we reproduce here:

We therefore treat the true surface region thickness and relative permittivity as material/process parameters that re-scale the surface loss tangents and thereby define the intrinsic loss factor that corresponds to p_{surf} as $\Gamma_{surf} = \sum_{k=SA,MS,MA} \frac{p_k}{p_{surf}} \frac{t_{k0}}{t_{surf}} \frac{\epsilon_{r0}}{\epsilon_r} \tan\delta_k$, where $\tan\delta_k$, t_{k0} , and ϵ_{r0} are the true dielectric loss tangent, thickness of the surface regions, and true dielectric constant, respectively [33,45].

We define a combined surface participation term, $p_{surf} = p_{SA} + p_{MS} + p_{MA}$ and define the corresponding surface loss factor as a weighted sum of the SA, MS, and MA loss factors (Supplementary Fig. S3). This construction of surface participation prevents us from distinguishing between the different surface losses, but because the relative scaling of these participations is roughly the same for all resonator and transmon geometries in this architecture, the geometric ratio p_k/p_{surf} is geometry-independent; therefore, Γ_{surf} still carries predictive power to estimate the loss of a desired resonator geometry.

Because the relative proportions of the surface participations are geometry independent in the coaxial architecture, the weighted sum of loss tangents that form the weighted surface loss factor is therefore roughly the same for different modes. The reviewer is correct to point out that differences in the weights p_k/p_{surf} between different modes can introduce systematic error in the extracted value of Γ_{surf} , since the effective Γ_{surf} would be slightly different for each mode. However, because p_k/p_{surf} is roughly the same for all the modes, the systematic error is not significant relative to the aforementioned device-to-device variation which is used to determine predicted ranges for device Qs. We have added a sentence in the main text in lines 201-206 to highlight this as well.

We have additionally added Supplementary Section 11, where we estimate the systematic error due to small variations in p_k/p_{surf} to be as little as 4% and as large as 11%, which is far smaller than the observed device-to-device variation of 20-40%. Since errors add in quadrature, the systematic errors become far smaller than the device-to-device variation and are therefore subdominant.

3. In Fig. 2d, Ta films on unannealed substrates tend to exhibit smaller surface losses compared to those on annealed substrates. This is confusing! In Tab. S1, the surface loss and bulk loss data are exactly the same for Ta on EFG-sapphire substrates with and without annealing. This mistake needs to be corrected for me to understand Fig. 2d.

We thank the reviewer for noticing this and bringing this to our attention. This was an obvious mistake that was overlooked, and we sincerely apologize for the confusion it has caused. We have corrected the mistake in Table S1, and have highlighted the changes in red. We have also reproduced the corrected table below:

Table S1: Summary of loss characterization devices

Device ID	Sapphire Growth Method	Anneal	Thin-Film Superconductor	Mode $Q_{\text{int}}(\bar{n} = 1)$ ($\times 10^6$)			Loss Factors ($\bar{n} = 1$)		
				D1	D2	C	Γ_{surf} ($\times 10^{-4}$)	Γ_{bulk} ($\times 10^{-8}$)	$1/g_{\text{scam}}$ (Ωm) ($\times 10^{-5}$)
AM22 TSL1	HEM	No	Al	0.55	8.95	9.00	14.5 + 1.0	7.96 + 0.5	13.4 + 0.9
AM22 TSL2				0.41	8.79	4.30	19.7 + 3.2	5.65 + 1.5	51.6 + 2.4
AM22 TSL3				0.39	7.13	6.41	20.8 + 6.3	8.78 + 3.0	22.4 + 1.2
DZ22 TSL3	HEM	Yes	Al	0.56	9.25	3.13	14.2 + 3.5	7.42 + 1.7	81.2 + 1.2
DZ22 TSL4				0.82	5.41	6.81	8.58 + 0.5	20.8 + 0.4	13.1 + 1.1
A23A1 TSL2	HEMEX	Yes	Al	1.53	10.33	12.21	6.83 + 0.4	4.14 + 1.8	1.95 + 1.6
A23A1 TSL3				0.88	7.56	10.17	12.3 + 0.5	2.57 + 0.7	1.56 + 0.3
EF21 TSL1	EFG	No	Ta	2.41	6.24	10.21	1.96 + 0.2	19.8 + 0.7	0.87 + 0.7
EF21 TSL2				1.91	6.22	9.93	2.89 + 0.1	19.5 + 1.2	1.26 + 1.2
EF21 TSL3				1.58	3.66	0.81	2.72 + 0.2	33.7 + 0.2	186 + 1.0
EF21 TSL4				1.63	3.65	0.48	2.54 + 0.3	33.4 + 2.0	331 + 1.5
EC21 ASL1	EFG	Yes	Ta	2.67 ¹	-	5.04	1.76 + 0.4	28.5 + 0.7	-
EC21 ASL2				3.41 ¹	-	10.99	3.42 + 0.2	7.16 + 0.7	-
EC21 ASL3				3.19 ¹	-	10.16	4.40 + 0.2	2.22 + 1.0	-
EC21 ASL4				2.14 ¹	-	15.38	6.77 + 0.2	1.62 + 0.8	-
R22 TSL1	HEM	Yes	Ta	1.59	24.04	24.20	5.00 + 1.0	2.78 + 0.5	1.08 + 0.4
R22 TSL3				2.04	26.01	7.06	3.87 + 0.2	2.83 + 0.2	11.9 + 1.4
R22 TSL4				1.78	19.66	13.69	4.35 + 0.6	4.23 + 0.3	3.92 + 0.3
BF22 TSL1	HEMEX	Yes	Ta	1.24	19.57	14.25	6.44 + 0.7	3.30 + 0.5	2.38 + 0.2
BF22 TSL2				2.53	27.40	14.96	3.07 + 0.1	2.95 + 0.1	2.81 + 0.2
BF22 TSL3				2.38	41.23	10.80	3.40 + 0.3	1.25 + 0.2	5.18 + 0.2
BF22 TSL4				2.44	29.47	16.54	3.22 + 0.1	2.56 + 0.2	2.45 + 0.3

¹ASL differential (D) mode Q

TSL: Tripole stripline

ASL: Adjacent stripline

4. In Figs. 3b and 5b, how were the predictions on the lifetimes of transmon qubit and quantum memory made? According to the supplementary tables, values of the loss factors in the same category vary significantly from device to device. Calculations based on the average of few scattered samples are always not that trustable. Without fully understanding the errors of the listed values in the supplementary tables for the abovementioned reasons, I think the claim "With this knowledge we predict and experimentally verify the relaxation times of aluminum- and tantalum-based transmon qubits" should be softened in tone.

The predictions were made using participation ratios calculated by finite element simulation and average surface and bulk loss factors calculated in Table S3. The surface loss factors here are categorized by the superconductor used (Al vs Ta) and substrate treatment (annealed vs unannealed). The bulk loss factors are likewise categorized by sapphire grade and substrate treatment. While we agree that averages based on a few scattered samples may not faithfully capture the true distribution of measurement outcomes, we believe that because the transmons and hairpin striplines were co-fabricated with the tripole striplines, the captured device-to-device variation was enough to make useful predictions of device performance, as demonstrated in Figs. 4b and 6a, which shows that we were able to make predictions, and experimentally verify those predictions using different materials, processes, and devices (Al and Ta processes, transmons and hairpin striplines).

5. I am puzzled why the x-indices in Figs. 3b and 5b were listed differently.

Figs. 3b and 5b show predicted losses for transmons and hairpin stripline quantum memories made using different superconductors and different substrates. The x-indices in Fig. 3b were chosen to reflect the materials systems used for the transmons that were actually measured in Fig. 4b. The inclusion of a loss prediction for Al transmons on annealed HEM sapphire was done so in order to illustrate that for our liftoff Al process, the relaxation rate is expected to be dominated by surface loss irrespective of substrate grade. In Fig 5b, we intended to highlight the fact that despite the optimized design, using just Al or just Ta yields modest coherence improvements; and annealing the substrate results in further improvements. However, both Ta and annealed HEMEX sapphire have to be adopted in order to achieve coherence times beyond a millisecond. This illustrates that improvements in both surface loss and bulk dielectric loss are important to achieve the coherences presented in this work.

REVIEWERS' COMMENTS

Reviewer #2 (Remarks to the Author):

I thank the authors for their detailed response which addresses most of my concerns. As I mentioned in my previous report that fighting decoherence induced by dielectric two-level states is a crucial subject for improving the ultimate performance of superconducting qubit devices, I think the authors have done a timely work with valuable output, which is suitable for publication in Nature Communications.

However, I still have a few technical comments:

1. I have a concern on Fig. 2b, as I do see a downward bending of the bulk loss when the photon number drops below 100. Typically, any kind of losses should increase or saturate with decreasing driving powers. Is this contradiction associated with some sort of systematic errors of their analysis, since the bulk loss factor is much smaller than the other two? I hope the authors can discuss a bit on this point.

2. Since the three curves in Fig. 2b are well separated along the y-axis, I recommend plotting them separately in three small panels, all in linear scale, so that I might see error bars. The current plotting style in logarithmic scale is not very informative. BTW, can the authors provide the fitted beta values in supplementary?

3. Again, the statement in the abstract that "With this knowledge we predict and experimentally verify the relaxation times of aluminum- and tantalum-based transmon qubits" needs to be softened in tone. If I understand correctly, the predicted lifetimes in Figs. 3b and 5b all have significant values in standard deviation (S.D.), so that overlaps between predictions for different configurations are obvious. For example, for quantum memory in Fig. 5b, the lower bound for Ta on HEMEX (A) is much lower than the upper bound for Ta on HEM (A) and also lower than that for Al on HEMEX (A). It should be noted that the confidence level for the interval defined by ± 1 S.D. is only 68%.

My suggestion for rephrasing: With this knowledge we predict the relaxation times of aluminum- and tantalum-based transmon qubits, and obtain experimental results that are in qualitative agreement.

4. In Table S2, the authors exclude samples with $\text{MRD} > 3$. How is the criterion 3 determined? The authors should clarify this point.

Reviewer #2 (Remarks to the Author):

Reviewer's comments in black; author's comments in red

I thank the authors for their detailed response which addresses most of my concerns. As I mentioned in my previous report that fighting decoherence induced by dielectric two-level states is a crucial subject for improving the ultimate performance of superconducting qubit devices, I think the authors have done a timely work with valuable output, which is suitable for publication in Nature Communications.

However, I still have a few technical comments:

1. I have a concern on Fig. 2b, as I do see a downward bending of the bulk loss when the photon number drops below 100. Typically, any kind of losses should increase or saturate with decreasing driving powers. Is this contradiction associated with some sort of systematic errors of their analysis, since the bulk loss factor is much smaller than the other two? I hope the authors can discuss a bit on this point.

To address this concern, we refer to the response to comment #2. We have plotted the curves in Fig 2b on a linear scale, and observe that the bulk loss factor increases at intermediate photon numbers (100-10000). The cause of this behavior is not quite clear, but we hypothesize that it may be related to spatial inhomogeneities in TLS saturation. The loss analysis presented in this work assumes that loss factors are locally homogenous for each device (for example, the bulk loss factor is the same throughout the volume of the chip on which a particular tripole stripline device is fabricated). Since the electromagnetic fields of each mode are spatially distributed throughout the chip in different ways, this assumption ensures that each mode is subjected to the same set of intrinsic loss factors. However, if there is inhomogeneity within a single chip, each mode may be subjected to a slightly different set of loss factors with different TLS saturation properties, leading to non-monotonic behavior of the extracted loss factor. This is a systematic error noticeable in the bulk loss factor due to it being much smaller (and therefore more sensitive to these inhomogeneities) than the other two. The behavior would also be most noticeable in the intermediate photon number regime, where the mode's Q is sensitive to changes in photon number.

We would like to highlight two things regarding this observation. Firstly, the bulk loss factor varies by a relatively small amount, around 20%; whereas the surface loss factor varies by a factor of 30. As a result, we can still call the bulk and seam loss factors relatively power independent, if not nearly power independent. Secondly, the variation in the bulk loss factor is further evidence of spatial inhomogeneity of loss factors, which is more directly interrogated by extracting the loss factors for multiple nominally identical as shown in Figs 2d and 2e. With this method, the device-to-device variation in bulk loss factor was found to be over 30% and captures both the aforementioned systematic error and the intrinsic spatial inhomogeneity. This is the number that is used in the calculation of the ranges of predicted Q for transmons and quantum memories.

We have added a short explanation in the caption of figure 2 that acknowledges this.

2. Since the three curves in Fig. 2b are well separated along the y-axis, I recommend plotting them separately in three small panels, all in linear scale, so that I might see error bars. The current plotting style in logarithmic scale is not very informative. BTW, can the authors provide the fitted beta values in supplementary?

We have altered Fig 2b according to the reviewer's recommendations, and we have added the TLS fit parameters, including the fitted beta values, in Table S13. Here, since the surface loss factor varies by over an order of magnitude and the fit error for this device is around 3%, the error bar is still difficult to see due to the relatively large span of the y axis. However, the error bars for the bulk dielectric and seam loss factors (~10%) are more clearly visible due to their relative power independence and small y axis span.

Table S13: TLS fit parameters

Device ID	D1 Mode				D2 Mode				C Mode			
	Q_0	$\tan \delta_{\text{TLS}}$	n_c	β	Q_0	$\tan \delta_{\text{TLS}}$	n_c	β	Q_0	$\tan \delta_{\text{TLS}}$	n_c	β
AM22 TSL1	7.1×10^6	2.9×10^{-3}	0.1	0.50	4.6×10^7	3.1×10^{-3}	28	0.22	2.1×10^7	4.1×10^{-3}	1.1	0.22
AM22 TSL2	4.2×10^6	3.6×10^{-3}	0.1	0.47	3.3×10^7	2.7×10^{-3}	140	0.26	6.2×10^6	3.7×10^{-3}	12	0.44
AM22 TSL3	2.2×10^6	8.2×10^{-2}	1.0×10^{-8}	0.42	1.8×10^7	2.7×10^{-3}	200	0.27	8.7×10^6	1.9×10^{-3}	4700	0.64
DZ22 TSL3	2.5×10^7	2.2×10^{-3}	0.6	0.59	2.0×10^7	2.4×10^{-3}	0.7	0.29	3.8×10^6	2.9×10^{-3}	6.0	0.55
DZ22 TSL4	4.2×10^7	1.1×10^{-3}	5.6	0.88	7.1×10^6	1.3×10^{-3}	120	0.61	9.4×10^6	2.1×10^{-3}	45	0.30
A23A1 TSL2	1.0×10^8	7.9×10^{-4}	6.8	0.81	2.0×10^8	1.4×10^{-3}	1.2	0.44	1.3×10^8	2.0×10^{-3}	0.9	0.35
A23A1 TSL3	7.0×10^7	1.8×10^{-3}	0.8	0.76	7.8×10^7	1.9×10^{-3}	1.0	0.52	3.0×10^7	1.8×10^{-3}	1.0	0.48
EF21 TSL1	4.0×10^7	3.7×10^{-4}	9.0	0.55	1.8×10^{14}	4.7×10^{-3}	2.7×10^6	0.65	2.5×10^7	2.8×10^{-3}	5.6×10^4	0.70
EF21 TSL2	2.8×10^7	4.9×10^{-4}	4.2	0.53	2.4×10^{13}	4.7×10^{-3}	4.4×10^5	0.59	1.0×10^{14}	4.8×10^{-3}	8.1×10	0.39
EF21 TSL3	5.2×10^7	6.3×10^{-4}	4.7	0.47	3.9×10^7	7.3×10^{-3}	1.9×10^6	0.71	8.5×10^5	2.9×10^{-3}	6.0×10^4	0.55
EF21 TSL4 ¹	2.5×10^7	8.1×10^{-4}	0.3	0.48	2.1×10^{13}	8.1×10^{-3}	6.3×10^5	0.62	4.8×10^5	N/A	N/A	N/A
EC21 ASL1 ²	2.0×10^7	7.6×10^{-4}	0.5	0.56	-	-	-	-	5.5×10^6	5.6×10^{-4}	430	0.76
EC21 ASL2 ²	1.4×10^7	3.6×10^{-4}	26	0.60	-	-	-	-	1.3×10^7	4.5×10^{-4}	280	0.61
EC21 ASL3 ²	1.9×10^7	4.2×10^{-4}	33	0.53	-	-	-	-	1.2×10^7	6.9×10^{-4}	38	0.63
EC21 ASL4 ²	1.2×10^7	6.8×10^{-4}	3.0	0.74	-	-	-	-	2.3×10^7	8.8×10^{-4}	48	0.34
R22 TSL1	1.5×10^7	6.8×10^{-4}	0.8	0.72	1.1×10^8	1.2×10^{-3}	4.7	0.29	1.0×10^8	1.8×10^{-3}	78	0.15
R22 TSL3	1.8×10^7	3.8×10^{-4}	13	0.83	5.8×10^7	6.4×10^{-4}	400	0.47	9.0×10^6	1.8×10^{-3}	380	0.12
R22 TSL4	1.0×10^8	6.3×10^{-4}	1.4	0.50	8.0×10^7	1.7×10^{-3}	0.2	0.21	1.7×10^7	6.5×10^{-4}	6600	0.43
BF22 TSL1	2.7×10^6	1.1×10^{-3}	1.1×10^{-3}	0.32	3.1×10^7	5.8×10^{-4}	200	0.37	2.0×10^7	1.4×10^{-3}	0.2	0.17
BF22 TSL2	1.0×10^8	4.0×10^{-4}	4.5	0.43	6.0×10^7	6.1×10^{-4}	280	0.33	2.1×10^7	1.2×10^{-3}	0.8	0.21
BF22 TSL3	9.5×10^{13}	3.7×10^{-4}	160	0.46	7.4×10^7	3.1×10^{-4}	1700	0.60	1.2×10^7	5.2×10^{-4}	8800	0.49
BF22 TSL4	1.6×10^8	4.3×10^{-4}	2.8	0.44	4.9×10^7	4.0×10^{-4}	2600	0.50	2.5×10^7	1.3×10^{-3}	1.7	0.15

¹EF21 TSL4 common mode was found to be power independent and the best interpolating function was found to be a constant value of Q_0 .

²ASL has only a differential (D) mode and a common (C) mode

TSL: Tripole stripline
ASL: Adjacent stripline

3. Again, the statement in the abstract that "With this knowledge we predict and experimentally verify the relaxation times of aluminum- and tantalum-based transmon qubits" needs to be softened in tone. If I understand correctly, the predicted lifetimes in Figs. 3b and 5b all have significant values in standard deviation (S.D.), so that overlaps between predictions for different configurations are obvious. For example, for quantum memory in Fig. 5b, the lower bound for Ta on HEMEX (A) is much lower than the upper bound for Ta on HEM (A) and also lower than that for Al on HEMEX (A). It should be noted that the confidence level for the interval defined by +/-1 S.D. is only 68%.

We have rephrased the statement to the following: "With this knowledge we predict the relaxation times of aluminum- and tantalum-based transmon qubits, and find that they are consistent with experimental results." We feel that this new phrasing appropriately softens the tone while still recognizing that the predictive model is consistent with experimental measurements for devices of vastly varying geometry (transmons and hairpin stripline quantum memories) and varying materials systems (Al vs Ta transmons).

4. In Table S2, the authors exclude samples with $MRD > 3$. How is the criterion 3 determined? The authors should clarify this point.

The MRD criterion was used to identify and exclude outliers which can heavily influence the mean and standard deviation when sample sizes are small (<10 in the case of this work). The choice of $MRD > 3$ designates outliers as values that are > 4 standard deviations away from the mean. We have added a statement in the caption of Table S2 to clarify.